# NBR1-mediated selective autophagy of ARF7 modulates root branching

Elise Ebstrup [1,5], Jeppe Ansbøl [1,5], Ana Paez-Garcia [2], Henry Culp [1], Jonathan Chevalier[1], Pauline Clemmens [1], Núria S Coll [3,4], Miguel A Moreno-Risueno [2] & Eleazar Rodriguez [1✉]

## Abstract

Auxin dictates root architecture via the Auxin Response Factor (ARF) family of transcription factors, which control lateral root (LR) formation. In *Arabidopsis*, ARF7 regulates the specification of prebranch sites (PBS) generating LRs through gene expression oscillations and plays a pivotal role during LR initiation. Despite the importance of ARF7 in this process, there is a surprising lack of knowledge about how ARF7 turnover is regulated and how this impacts root architecture. Here, we show that ARF7 accumulates in autophagy mutants and is degraded through NBR1-dependent selective autophagy. We demonstrate that the previously reported rhythmic changes to ARF7 abundance in roots are modulated via autophagy and might occur in other tissues. In addition, we show that the level of co-localization between ARF7 and autophagy markers oscillates and can be modulated by auxin to trigger ARF7 turnover. Furthermore, we observe that autophagy impairment prevents ARF7 oscillation and reduces both PBS establishment and LR formation. In conclusion, we report a novel role for autophagy during development, namely by enacting auxin-induced selective degradation of ARF7 to optimize periodic root branching.

**Keywords** ARF7; Autophagy; Auxin; Lateral Root; NBR1
**Subject Categories** Autophagy & Cell Death; Plant Biology; Signal Transduction

## Introduction

The plant root system plays a fundamental role as a point of contact with the plant surroundings and allows adaptation to multiple stresses. Contrary to primary roots, lateral root (LR) development is postembryonic, which allows LR architecture to be influenced by growth conditions (Lv et al, 2021). As LR growth is modular, perception of the correct developmental conditions enables the repetition of LR modules along the primary root, maximizing the

root's surface area. LRs originate from a cell type called pericycle which surrounds the vasculature, and occurs sequentially in four steps: positioning, initiation, outgrowth, and emergence. During the positioning step, the plant ensures the spatiotemporal distribution, priming and activation of LR founder cells (LRFC) (Du and Scheres, 2018). LR spacing is controlled by the root clock which dictates oscillatory gene expression and cycles every 4–6 h (Moreno-Risueño et al, 2010). This oscillatory clock is regulated, among others, by the phytohormone auxin and can be characterized through monitoring of reporter marker expression driven by DR5, a synthetic promoter containing tandem repeats of an auxin-responsive TGTCTC element (Ulmasov et al, 1997). DR5 expression analyses led to the definition of two contrasting oscillation phases: in-phase (concurrent with DR5 maximum signal) and anti-phase (DR5 minimum signal) which might reciprocally repress each other (Moreno-Risueño et al, 2010). One of the genes displaying anti-phase expression is AUXIN-RESPONSIVE FACTOR 7 (*ARF7*), a key transcription factor (TF) which, among others, regulates different steps of LR formation (Okushima et al, 2007). Importantly, analysis of *ARF7* loss-of-function mutant (Moreno-Risueño et al, 2010; Perianez-Rodriguez et al, 2021) indicates that ARF7 is part of the core oscillator of the root clock and that this TF's oscillation is essential for the correct establishment of LR prebranch sites (PBS).

While positioning and initiation are broadly regulated by different mechanisms (Wachsman et al, 2020; Perez-Garcia et al, 2022), ARF7 and its two direct targets LATERAL ORGAN BOUNDARIES DOMAIN 16 (LBD16) and LBD33 are among the few genes that participate in both processes (Moreno-Risueño et al, 2010) which attests to ARF7 pivotal role in organizing root architecture. During the initiation step, which follows PBS specification, the auxin signaling node IAA14-ARF7-ARF19 controls LBD16 activation in the LRFC, which is necessary to achieve asymmetry and initiate cell division (Goh et al, 2019). In addition to the importance of *ARF7* transcriptional oscillation for LR formation, it has also been shown that the subcellular localization of ARF7 is important for its function: nuclear localization correlates with auxin-responsive tissues while in less responsive areas, activating ARFs concentrate to form cytoplasmic condensates with liquid-like and solid-like proprieties (Powers et al, 2019). This body of evidence demonstrates the regulatory

[1]Department of Biology, University of Copenhagen, 2200 Copenhagen N, Denmark. [2]Centro de Biotecnología y Genómica de Plantas (Universidad Politécnica de Madrid (UPM)—Instituto Nacional de Investigación y Tecnología Agraria y Alimentaria—CSIC (INIA/CSIC)). Campus de Montegancedo, Pozuelo de Alarcón, 28223 Madrid, Spain. [3]Centre for Research in Agricultural Genomics (CRAG), CSIC-IRTA-UAB-UB, Bellaterra 08193, Spain. [4]Consejo Superior de Investigaciones Científicas (CSIC), Barcelona 08001, Spain. [5]These authors contributed equally: Elise Ebstrup, Jeppe Ansbøl. ✉E-mail: Eleazar.rodriguez@bio.ku.dk

complexity coordinating ARF-dependent responses, encompassing an intricate web of processes which tightly regulate ARFs levels and compartmentalization. Yet surprisingly, little is known about how ARF7 degradation is managed and how this impacts the cellular decision-making behind LR formation.

Macroautophagy (herein autophagy) is an evolutionary conserved catabolic process which partakes in cellular recycling of macromolecules and organelles (Levine and Klionsky, 2004). Autophagy is characterized by the formation of an isolation membrane enclosing a portion of cytoplasm to form a double-membraned vesicle termed the autophagosome. Originally characterized during starvation stress, autophagy was viewed as a bulk degradation process due to the observation of what seemed to be an indiscriminate engulfment of portions of the cytoplasm. Further studies unveiled the existence of several selective cargo adapters which allow for targeted degradation of cellular constituents (e.g., Svenning et al, 2011; Nolan et al, 2017; Stephani et al, 2020; Wu et al, 2021), meaning that autophagy can be highly selective and participate in targeted remodeling of the cell. Perhaps one of the most characterized cargo receptors in plants is NEIGHBOUR OF BRCA1 (NBR1, Svenning et al, 2011), a hybrid of mammalian NBR1 and p62, which possess a PB1 domain necessary for homopolymerization and biomolecular condensate formation, and UBA domains for ubiquitin binding. It is through the latter that NBR1 binds ubiquitinated cargo, which is then targeted for autophagic degradation via NBR1 interaction with the autophagosome decorating proteins ATG8s (e.g., Zhou et al, 2013, 2014; Hafrén et al, 2017; Ji et al, 2020; Tarnowski et al, 2020). The ability to function selectively allows autophagy to participate in a variety of stress responses, namely by promoting the cellular reprogramming necessary to enact appropriate responses (Rodriguez et al, 2020). Consistent with this homeostatic role, autophagic deficiency has been shown to cause diverse developmental defects in animals; interference with core autophagic machinery component often causes lethality during early developmental stages, severe developmental defects, and shortened life span (e.g., Scott et al, 2004; Liu et al, 2015; Yoshii et al, 2016). In contrast, plants are able to tolerate disruption of autophagy activity without major penalties: autophagy-deficient mutants display growth phenotypes such as decrease in biomass, reduced inflorescence numbers and premature senescence (Bassham et al, 2006), but importantly, they are able to complete their life cycle and generate fertile progeny. However, these mild phenotypes displayed by plant atg mutants become more apparent under stress, as can be attested by the many publications linking autophagy with stress responses (Liu et al, 2009; Hofius et al, 2009; Vanhee et al, 2011; Zhou et al, 2013; Coll et al, 2014; Munch et al, 2014; Chen et al, 2015; Nolan et al, 2017; Sedaghatmehr et al, 2019; Rodriguez et al, 2020). For instance, under nutrient starvation, autophagy has been shown to help sustain primary root growth and LR formation (Deb et al, 2014; Huang et al, 2019). However, if and how autophagy impacts root architecture under normal developmental conditions is currently unknown and warrants close examination.

In this work, we explored how autophagy impacts ARF7-dependent responses during LR formation. By using a combination of cell imaging, biochemical, genetics, molecular biology, and proteomics approaches, we demonstrate that ARF7 turnover is processed via NBR1-mediated selective autophagy. Importantly, we demonstrate that ARF7 abundance, which rhythmically oscillates,

requires autophagy, and subsequently autophagy alters the oscillatory patterning. Moreover, we show that a short auxin treatment is sufficient to trigger ARF7 condensation and co-localization with ATG8a, though this mainly occurs in the oscillation and maturation zones. Consistent with this, autophagy deficiency reduces PBS establishment and LR formation. With these findings, we reveal an important regulatory role for selective autophagy during auxin responses through the turnover of the crucial auxin signaling regulator ARF7 and consequent fine-tuning of root architecture.

## Results

### Autophagy modulates ARFs turnover

We have recently reported that autophagy participates in stem cell reprogramming and differentiation in the distantly related species *Physcomitrium patens* and *Arabidopsis* (Rodriguez et al, 2020; Kanne et al, 2021). Because *Arabidopsis* stem cell hormonal reprogramming executes a developmental program similar to LR initiation (Sugimoto et al, 2010), we wondered if key regulators of these processes might be modulated via autophagy. As seen in Fig. 1, ARF7 accumulates in different autophagy mutants (atg) and in the knock-out mutant (KO) of the autophagic cargo adapter NBR1 (Fig. 1A), when compared to Col-0. To address if the proteasome also contributes to ARF7 proteostasis, we treated seedlings of those genotypes with the proteasome inhibitor MG132 (Fig. 1A). Col-0 samples treated with MG132 showed ARF7 accumulation to levels comparable with those of untreated atg mutants. Moreover, simultaneous inhibition of both degradation pathways (MG132 treatment in atg mutants) led to further accumulation of ARF7, which indicate that ARF7 degradation is dependent on autophagy and the proteasome and is in agreement with previous data showing proteasomal degradation of ARF7 (Jing et al, 2022). To rule out that elevated transcript levels in atg mutants could underscore ARF7 accumulation in those lines, we performed qPCR and observed that *ARF7* relative expression was not statistically significant among lines tested (Fig. 1B).

To further support our claim that autophagy modulates ARF7 turnover, we introduced the translational reporter *pARF7::gARF7-Venus* herein called *ARF7-Venus* (Orosa-Puente et al, 2018) into *atg2-1* background. We then quantified ARF7-Venus levels by western blot analyses of protein extracts from *ARF7-Venus* and *ARF7-Venus x atg2-1*, detecting significantly higher levels of ARF7-Venus when autophagy is dysfunctional (Fig. 1C). Using the same lines, we also analyzed ARF7-Venus subcellular fluorescence intensity in the root meristematic and maturation zones by confocal microscopy (Fig. 1D–I). In agreement with the western blot data, we detected significantly higher ARF7-Venus fluorescence intensity in the *atg2-1* background for all tissues and subcellular localization except for the meristem's nuclei (Fig. 1I). Because ARF7 often works together with ARF19 during regulation of root architecture, we also enquired about a role for autophagy during ARF19 turnover. Western blot probing of *pARF19::ARF19-GFP* expressed in an *ARF7* and *ARF19* double loss-of-function mutant (*nph4-1 arf19-1*) (Okushima et al, 2007) treated with or without the vacuolar protease inhibitors Pepstatin A and E-64D for 24 h revealed substantial accumulation of ARF19-GFP when vacuolar proteases were inhibited

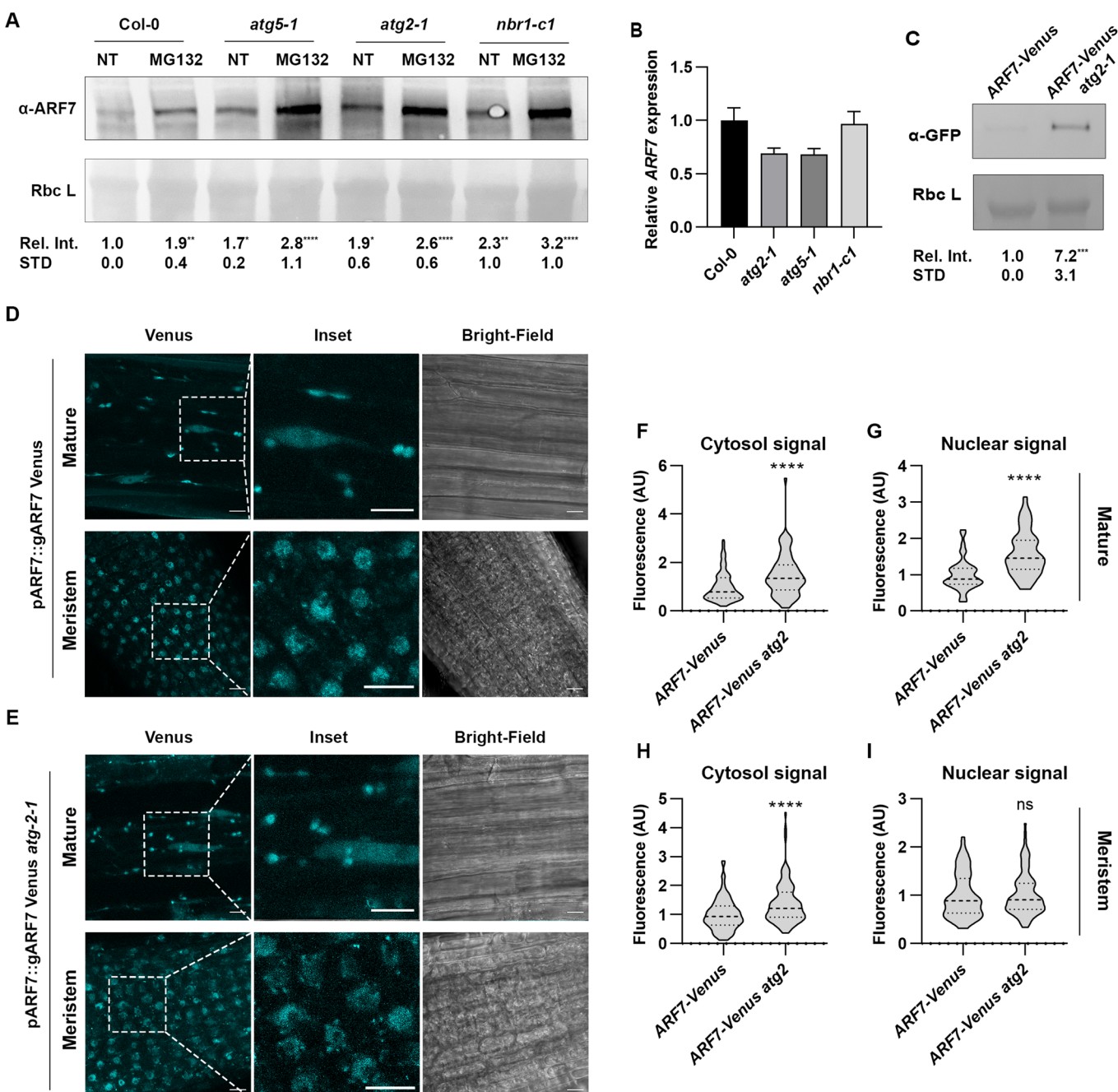

**Figure 1. ARF7 is degraded via autophagy.**

(A) ARF7 western blots of protein extracts from Col-0, atg5-1, atg2-1, and nbr1-c1 with or without the addition of the proteasome inhibitor MG132. (B) ARF7 relative expression in given genotypes. ARF7 expression was normalized to ACTIN2, and values are relative to Col-0 (set to 1). The experiment was repeated 3 times (biological replicates) with similar results; graph is a representative of one of the biological replicates. Error bars are representative of the standard deviation of the mean. (C) GFP western blot of protein extracts from Col-0 or atg2-1 expressing ARF7-Venus. Values below each band represent the relative intensity ratio between ARF7 (A) or ARF7-Venus (C) and RuBisCO large subunit (Rbc L) as stained with Ponceau S used as loading control. Ratios were normalized to untreated Col-0 NT (MS+ solvent) which was arbitrarily set to 1. Values provided are the average and SD of 5 (A) or 7 (C) biological replicates, asterisk depict statistical significance from Col-0 NT according to a Holm–Sidak test (A) or two-tailed t test (C) (*0.05; **0.01; ***0.001; ****0.0001). (D, E) Confocal images of root maturation zone (top panel) and meristematic zone (bottom panel) in Col-0 (D) or atg2-1 (E) expressing ARF7-Venus. Scale bar: 10 μM. (F–I) Fluorescence intensity quantification of ARF7-Venus subcellular localization in given lines. The values presented for ARF7-Venus fluorescence intensity in atg2-1 background in the maturation zone (F, G) or meristematic zone (H, I) were normalized to the values in Col-0 background (which was set to 1). The outline of the violin plots represents the probability of the kernel density. Dotted lines represent interquartile ranges (IQR), with the thick horizontal line representing the median; whiskers extend to the highest and lowest data point. Results were obtained from three independent experiments with at least 11 plants per condition being analyzed in total. At least 40 nuclei and cytoplasmic regions were measured per genotype/tissue. Asterisks mark statistical significance to NT according to a Mann–Whitney U test (****P < 0.0001). Source data are available online for this figure.

(Fig. EV1). Together, our results indicate that autophagy participates in the turnover of these essential developmental regulators.

## ARF7 interacts with NBR1 and ATG8a in planta

To further substantiate our claim that autophagy participates in the targeted degradation of ARF7, we performed protein-protein interaction studies. For this purpose, we used stable transgenic lines expressing ARF7-Venus and performed coimmunoprecipitation (Co-IP) experiments using anti-NBR1 and anti-ATG8a antibodies. The experiment revealed that NBR1 (Fig. 2A) but not GOLGI TRANSPORT 1 (GOT1) YFP, co-immunoprecipitated with ARF7-Venus; this is supported by an observable enrichment of ATG8a in the same conditions (Fig. 2B). Moreover, a web-based analysis identified several potential ATG8-interaction motifs (AIM) in ARF7 (Jacomin et al, 2016), supporting our findings. As a cautionary note, we were unable to detect ARF7-Venus in the input of both Co-IPs which can likely be explained by the fact that ARF7-Venus is under the control of its native promoter and thus lowly expressed and/or differences between the protein extraction buffers used. We also performed multi-color Bimolecular fluorescence complementation (mcBiFC) analysis with a system allowing simultaneous interaction studies of several proteins (Gehl et al, 2009) to transiently express Venus$^N$-ARF7 with SCFP$^N$-ATG8a and either SCFP$^C$-NBR1 or SCFP$^C$-TSPO, (TRYPTOPHAN-RICH SENSORY PROTEIN, an autophagic cargo adapter for aquaporins, Hachez et al, 2014) in *Nicotiana benthamiana* leaves. As expected, SCFP$^C$-NBR1 and SCFP$^N$-ATG8a were able to reconstitute CFP to produce distinct fluorescence signal localized in the nuclei and cytoplasm (Fig. 2C). Moreover, Venus$^N$-ARF7 and SCFP$^C$-NBR1 produced GFP signal, which was localized to the cytoplasm and nuclei, in many instances co-localizing with NBR1-ATG8a CFP signal (Fig. 2C). In contrast, SCFP$^C$-TSPO did not produce GFP signal when co-expressed with Venus$^N$-ARF7. In combination, these results indicate that ARF7 interacts with autophagic components and cumulative with data from Fig. 1, indicate that this TF is degraded, in part, via NBR1-mediated selective autophagy.

## ARF7 co-localizes with autophagosomes following and varies with time

Having established that autophagy regulates ARF7 abundance, it would be expected that this TF co-localized with autophagosome-like structures. To assess this, we crossed ARF7-Venus to mCherry-ATG8a and mCherry-NBR1 reporter lines (Svenning et al, 2011) and used this for live cell imaging. As expected, both autophagy-related proteins localized to cytoplasmic foci, in many instances co-localizing with ARF7-Venus (Fig. 3A). Moreover, mCherry-ATG8a also co-localized with ARF7-Venus in the nucleus (Fig. 3A). It should be noted that the co-localization of these autophagic proteins with ARF7 in cytoplasmic foci is mainly observable in mature tissue (Fig. 3A,B); in younger meristematic tissue ARF7 primarily localizes in the nucleus although some instances of cytoplasmic co-localization are also observable (Fig. 3A, white arrows). These findings are in agreement with previous reports demonstrating that ARF7 partitioning is dependent on the developmental stage of the tissue (Orosa-Puente et al, 2018; Powers et al, 2019).

ARF7's nucleo-cytopartitioning has been ascribed to liquid-liquid phase separation (LLPS) (Powers et al, 2019); likewise,

autophagosome formation sites and protein cargo triage (Kirkin et al, 2009; Zhang et al, 2009; Fujioka et al, 2020) are also dependent on LLPS. To address the nature of these cytoplasmic ARF7 & ATG8 foci, we incubated seedlings expressing mCherry-ATG8 and ARF7-Venus with the LLPS inhibitor 1,6-hexanediol (Wang et al, 2021; Xie et al, 2021) and tracked the stability of those foci over time. Most of the foci in which ARF7-Venus and mCherry-ATG8a co-localized were sensitive to 1,6-hexanediol treatment (Fig EV2), suggesting that these structures have liquid-like properties. Our data is both in agreement with findings for ARF7 (Powers et al, 2019) and studies in mammalians showing that NBR1 (and p62) is involved in condensate formation during autophagic degradation of substrates (Turco et al, 2021).

Interestingly, during the course of our experiments, we noticed that the co-localization of ARF7-Venus with mCherry-ATG8a varied with time, i.e., there was a significant increase (45–85%, $P < 0.05$) in ARF7 and ATG8a co-localization in the maturation zone when comparing seedlings during a period of ~6 h. This is consistent with previous reports (Perianez-Rodriguez et al, 2021) showing that *ARF7* transcript levels follow periodic fluctuation in roots. Next, we incubated seedlings with vacuole protease inhibitors (Pepstatin A and E-64d) which led to a dramatic increase in ARF7-Venus co-localization with mCherry-ATG8a (Fig. 3C,D). This both supports our findings with *ARF7-Venus* x *atg2-1* (Fig. 1) as well as the proposition that ARF7 condensate formation can be linked to autophagic degradation. Altogether, our results suggest that ARF7 cytoplasmic localization might be part of a proteome remodeling strategy which is mediated, at least in part, via autophagy.

## Autophagy regulates rhythmic changes to ARF7 protein abundance and PBS establishment

Intrigued by our results showing that co-localization between ARF7 and ATG8a follows rhythmic variation and knowing that *ARF7* expression and protein abundance also show a similar pattern in the oscillation zone (OZ, Perianez-Rodriguez et al, 2021), we decided to explore the impact of autophagy in the natural variation of ARF7's abundance. Therefore, we performed western blots against ARF7 in Col-0 seedlings over time (Figs. 4A and EV3) where it can be seen that ARF7 levels display an oscillatory pattern. Moreover, these results likely explain the variation in co-localization of ARF7-Venus with autophagosomes at different timepoints (Fig. 3): condensation is dependent on target concentration (Mcswiggen et al, 2019) and therefore more likely observable when ARF7 levels increase. Our findings also illustrate that ARF7 abundance display rhythmic changes that could affect plant tissues outside roots. This is consistent *ARF7*'s expression profile (e.g., Rosado et al, 2012; Lee et al, 2019) and with ARF7's role during hypocotyl gravi- and phototropism (Wang et al, 2020).

In marked contrast, the rhythmic variation in ARF7 abundance was lost ($P \leq 0.05$) when autophagy was disrupted (Fig. 4B–E). Moreover, the transcript levels of *LBD16*, which is directly regulated by ARF7, also lost its rhythmic fluctuation in *atg* mutants (Fig. 4F). Interestingly, *LBD16* levels are substantially higher in *atg2-1* than Col-0, which could be related to the overall higher levels of ARF7 seen in *atg* mutants. Taken together, these results demonstrate the importance of autophagy in regulating the ARF7-dependent rhythm.

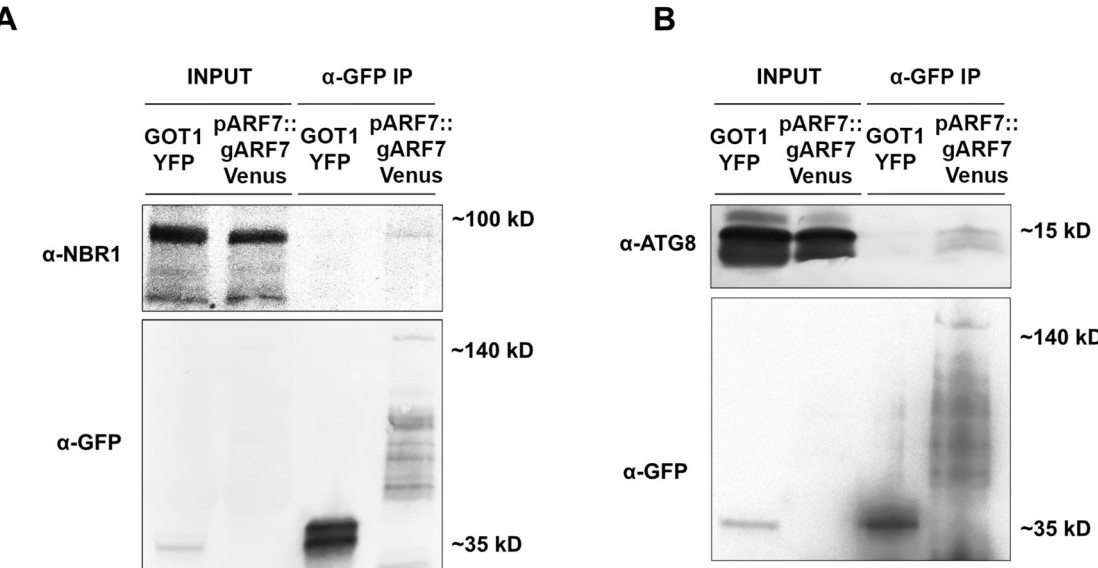

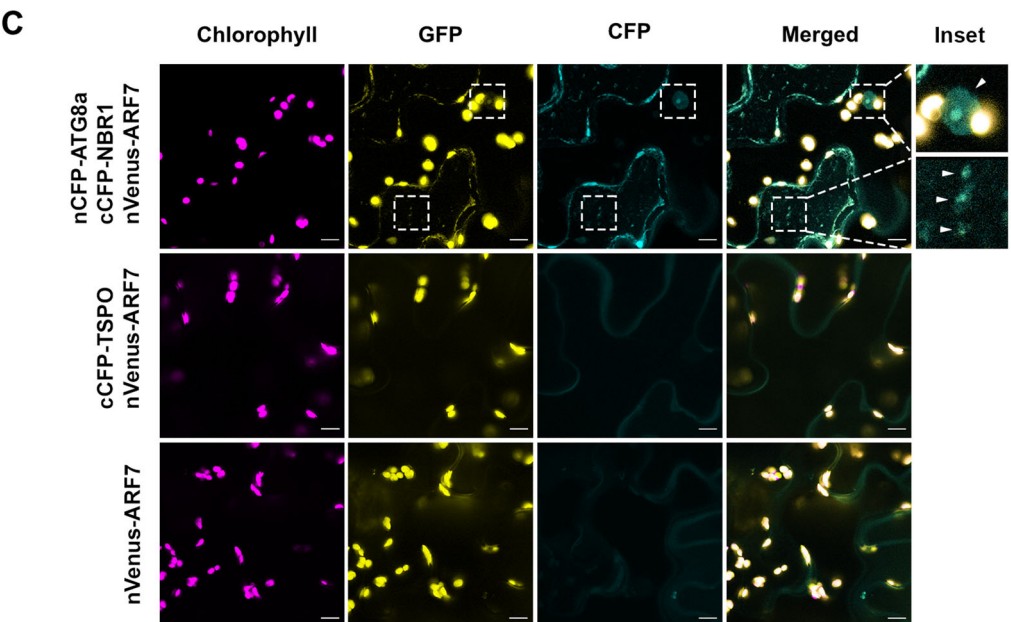

**Figure 2. ARF7 associates with the autophagic components NBR1 and ATG8a.**

(A, B) ARF7 and ATG8/NBR1 association was tested using a coimmunoprecipitation (Co-IP) assay. Protein extracted from *Arabidopsis* lines expressing *pARF7::gARF7-Venus* or free *YFP* were immunoprecipitated with anti-GFP antibody and subsequently probed with antibodies raised against NBR1 or ATG8a. Apparent molecular weights: ARF7-Venus ~150 kDa; GOT1-GFP ~45 kDa; NBR1-100 kDa; ATG8 ~15 kDa; (C) Bimolecular fluorescence complementation experiments showed oligomerization of ARF7, NBR1 and ATG8a in *N. Benthamiana*. Images were taken 48 hpi. cCFP-TSPO was used as control. Insets display close-ups of examples of reconstitution of GFP and CFP signal, which are also indicated by arrows. Etched boxes in GFP and CFP channels indicate the place of the insets in those channels. Scale bar: 10 µM Source data are available online for this figure.

*ARF7* oscillation has been established as an essential step during PBS determination as *arf7-1* mutants display abnormal PBS patterning (Perianez-Rodriguez et al, 2021). Because autophagy is necessary for ARF7 turnover (Figs. 1–3), we questioned whether PBS establishment could be affected when autophagy is impaired. To track this, we used a reporter line expressing *DR5::Luciferase*

and compared PBS numbers and the luminescence intensity in the OZ of seedlings after 7 days treatment with mock (DMSO) or Pepstatin A. As seen in Fig. 4G–I, seedlings treated with Pepstatin A had decreased number of PBS and OZ luminescence when compared to the mock-treated samples ($P \leq 0.05$). In addition, confocal visualization of ARF7-Venus levels in the OZ also show

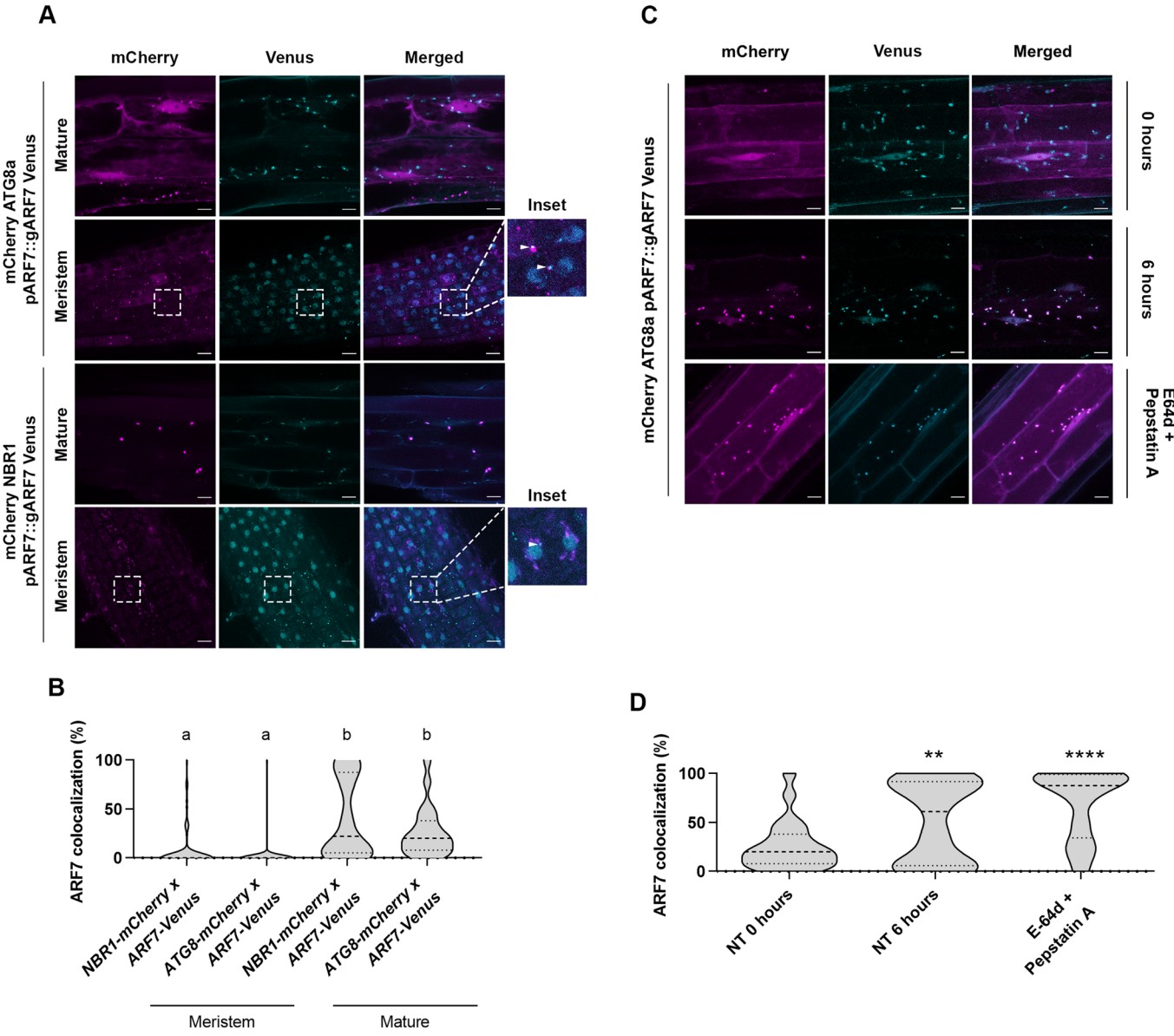

**Figure 3. ARF7 co-localization with autophagosomes displays spatial and rhythmic variation.**

(A) Co-localization of ARF7-Venus with mCherry-ATG8a (nuclei and/or cytoplasmic foci) or mCherry-NBR1 (cytoplasmic foci) in meristem or maturation zone. Pictures were taken in the respective root zone of 10–14-day old *Arabidopsis* seedlings. Scale bar 10 μM. Insets display close-ups of regions of the meristem, white arrow indicates instances of co-localization between mCherry-ATG8a (Top)/mCherry-NBR1(bottom) and ARF7-Venus foci. Etched boxes in mCherry and Venus channels indicate the place of the insets in those channels. (B) Quantification of the co-localization of ARF7-Venus with mCherry-ATG8a or NBR1 under conditions depicted in (A); samples followed by the same letters are not statistically different (Dunn's test, $P \leq 0.0001$). At least four plants were analyzed per genotype (C) ARF7-Venus co-localization with autophagosomes under selected treatments: No Treatment (NT, 0 h or 6 h), E-64d + Pepstatin A (vacuolar protease inhibitors. Scale bar 10 μM. (D) Quantification of the co-localization of ARF7-Venus with mCherry-ATG8a under conditions depicted in (C). The outline of the violin plots represents the probability of the kernel density. Dotted lines represent interquartile ranges (IQR), with the thick horizontal line representing the median; whiskers extend to the highest and lowest data point. Results were obtained from three independent experiments with at least eight plants per condition. Asterisks mark statistical significance to NT (MS+ solvents) according to a Wilcoxon-rank test (**0.01; ****0.0001). Source data are available online for this figure.

that these are significantly higher in *atg2-1* cytoplasm ($P \leq 0.005$) and nuclei ($P \leq 0.0005$) when compared to control (Fig. 4J–L). As ARF7 is a repressor of the in-phase oscillations (Perianez-Rodriguez et al, 2021), taken together, these results indicate that accumulation of ARF7 due to defective proteostasis leads to decreased *DR5* signal in the OZ and this in turn affects PBS establishment.

## Auxin-promoted ARF7 degradation requires functional autophagy

As the root clock oscillations are deeply connected to auxin, we wanted to explore if auxin treatment impacts ARF7 turnover. Consequently, we treated *ARF7-Venus x mCherry-ATG8a* seedlings with auxin (Fig. 5A,B) for 30 min as we have previously shown that

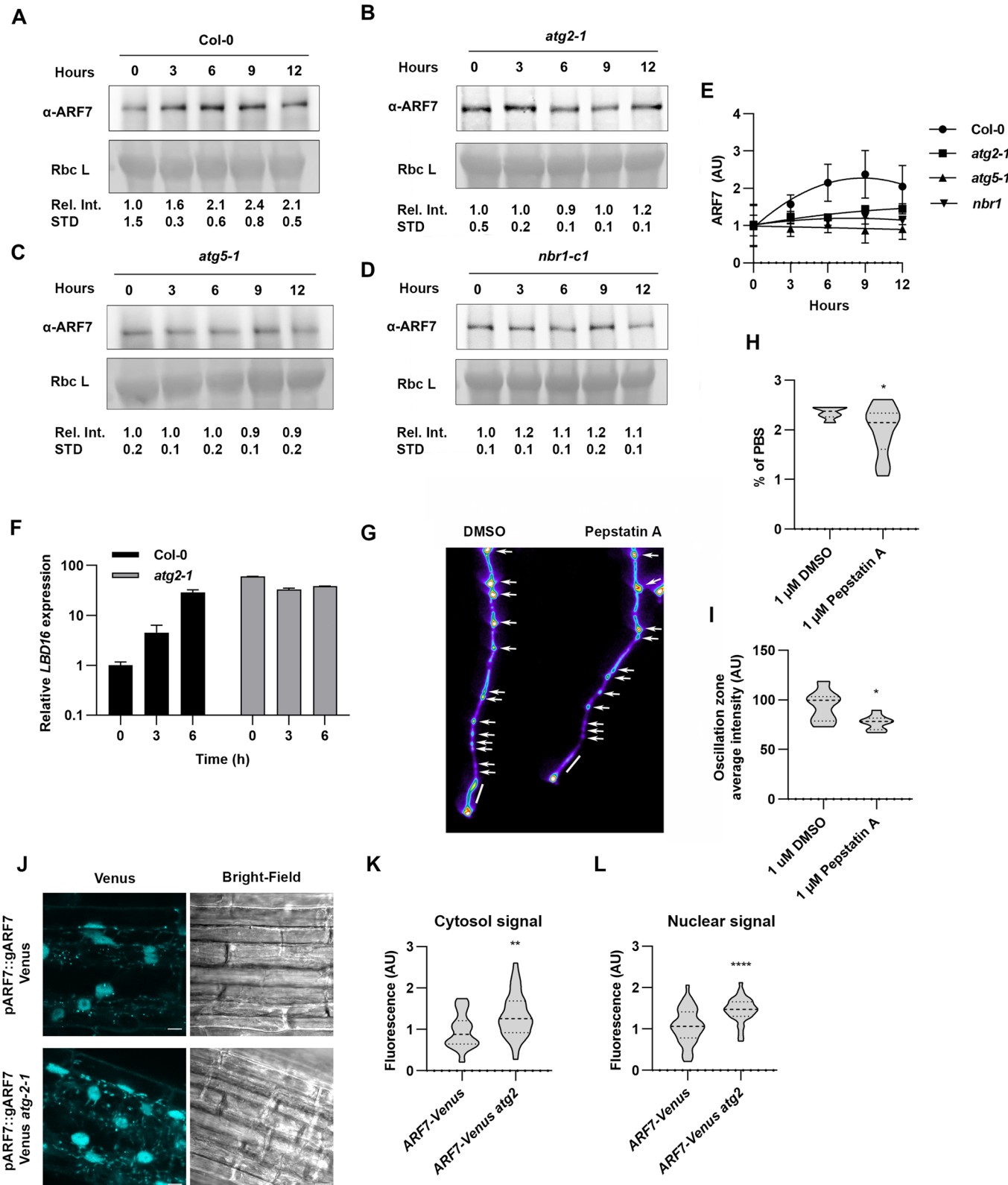

Figure 4.  ARF7 abundance displays rhythmic variation and is modulated by autophagy.

(A–D) ARF7 western blots from proteins extracted from Col-0 (A); *atg2-1* (B); *atg5-1* (C), or *nbr1-c1* (D) at given hours (0–12 h). Values below each band represent the relative intensity ratio between ARF7 and RuBisCO large subunit (Rbc L), using Ponceau S as loading control. Ratios were normalized to untreated (NT) which was arbitrarily set to 1. Values given are the mean ± SD of three independent experiments, Col-0 was statistically significant ($P \leq 0.05$) from all autophagy mutant, according to a Paired *T* test. (E) Correlation of the oscillatory pattern for ARF7 protein level over 12 h period in in Col-0 and *atg* mutants. (F) Relative gene expression of *LBD16* in Col-0 and *atg2-1* at given timepoints. *LBD16* expression was normalized to *ACTIN2* and set to 1 for each genotype at timepoint 0 h. Experiments was repeated three times (biological replicates) with similar results; graph is a representative of one of the biological replicates. (G) *DR5::Luciferase* images of mock (DMSO) or Pepstatin A (1 μM)-grown seedlings depicting PBS (white arrows) and OZ (white bar). (H, I) Quantification of (H) PBS (%) and (I) OZ luminescence intensity depicted from plants as treated in (G). The outline of the violin plots represents the probability of the kernel density. Dotted lines represent interquartile ranges (IQR), with the thick horizontal line representing the median; whiskers extend to the highest and lowest data point. Results represent values from at least nine plants per condition. Asterisks mark statistical significance to NT according to a Welch *t* test (*0.05). (J) Confocal imaging of ARF7-Venus or *atg2-1* ARF7-Venus in the OZ. Scale bar: 10 μM (K, L) Quantification of the fluorescence intensity in cytosol (K) or nuclei (L) under the conditions depicted in (H). The outline of the violin plots represents the probability of the kernel density. Dotted lines represent interquartile ranges (IQR), with the thick horizontal line representing the median; whiskers extend to the highest and lowest data point. Results were obtained from three independent experiments with at least eight plants per condition. Asterisks mark statistical significance to NT according to a Mann–Whitney test (**0.01; ****0.0001). Source data are available online for this figure.

autophagy is rapidly triggered upon auxin perception (Rodriguez et al, 2020). Remarkably, auxin treatment led to rapid and massive ARF7-Venus localization in cytosolic foci, for all tissues tested, indicating that auxin promotes ARF7 condensation. Likewise, we also observed a massive increase in mCherry-ATG8 foci (Fig. 5), which is consistent with our previous reports showing that quick bursts of auxin lead to a rapid increase in autophagic activity (Rodriguez et al, 2020). Surprisingly, these dramatic increases of ARF7 and ATG8a cytoplasmic foci only produced a modest, albeit significant, increase in co-localization in the meristem, which contrasts with the strong co-localization observed in the OZ and maturation zone (Fig. 5A,B). Given that: (a) autophagy has a well characterized role in aggregate degradation (e.g., Lim et al, 2015); (b) the root meristem has been shown to display zone-specific chaperonin expression to prevent protein aggregation (Llamas et al, 2021); (c) ARF7 condensates have liquid and solid properties (Fig. EV2, Powers et al, 2019); (d) ARF7 condensates are more prominent in older tissue (Fig. 3, Powers et al, 2019); our results here likely reveal a mechanism by which auxin treatment leads to a rapid activation of autophagy to modulate ARF7 levels though this mainly occurs in the oscillation and maturation zones, where aggregation is more prone to occur. We also tested whether cytokinin, a hormone often producing contrasting effects to auxin, lead to changes in co-localization between ARF7 and ATG8 (Fig. EV4), but this was not the case ($P > 0.05$).

Given that NAA induced rapid ARF7 localization to cytoplasmic foci, we wondered if this would have any effect in ARF7 abundance. Hence, we checked ARF7 levels by western blot with protein extracts from Col-0 and *atg2-1* seedlings treated with NAA for 1 h (Fig. 5C,D). Our data revealed that NAA produced a mild but significant decrease ($P \leq 0.01$) in ARF7 levels in Col-0, which is consistent with the NAA-induced increase co-localization between ARF7 and autophagic components (Fig. 5A,B). In combination, our results (Figs. 3–5) support a mechanism by which auxin promotes ARF7 condensation to modulate ARF7 levels, preventing excessive nuclear ARF7 and consequent exaggerated responses. In contrast to previous reports (e.g., Wang et al, 2005; Perianez-Rodriguez et al, 2021), our data suggests that auxin treatment induces certain changes to ARF7 abundance; which has also been reported at the transcript level after 2 h of IAA treatment (Biswas et al, 2019). These differences might be easily conciliated due to different experimental set ups among reports: longer auxin treatments (3h-16h), different tissues (i.e., root's OZ or protoplast) and type of auxin were used in those

reports (Perianez-Rodriguez et al, 2021; Wang et al, 2005) as opposed to the experimental setup used here. Concerning *atg2-1* seedlings, NAA treatment did not lead to significant changes in ARF7 levels (Fig. 5C,D) which might not be surprising given that ARF7 levels are already elevated when autophagy is impaired and that other compensatory mechanisms (e.g., proteasome) could be upregulated to compensate for defective autophagy. Accumulation of ARF7 in the cytoplasm, which we have shown is impacted by autophagy (Figs. 1D–I and 3), has been linked to decreased auxin-dependent responses (Powers et al, 2019). As *atg2-1* is unable to execute ARF7 turnover upon rapid auxin treatment (Fig. 5), we wondered if this could be indicative of more broad implications, namely altered proteome remodeling during rapid auxin enacted responses. To evaluate this, we gave a short auxin treatment (30 min) to Col-0 and *atg2-1* seedlings and performed Mass Spectrometry (MS). Despite the rapid nature of the treatment with NAA, proteome remodeling was engaged in both genotypes (Fig. 5E), and as expected, more proteins were downregulated in Col-0 than *atg2-1* (167 vs 94, Fig. 5F). This difference in auxin-induced protein downregulation between genotypes is consistent with the loss of a major proteostasis pathway in the *atg2-1* background. There were also differences, albeit smaller, in what concerns auxin-induced protein accumulation, 146 for Col-0 vs 114 for *atg2-1*. Surprisingly, only around 5% of the proteins with differential regulation upon NAA treatment were similarly down (or up) regulated in both genotypes. These results indicate that while *atg2-1* responds to NAA, the nature of this response is substantially different to that of Col-0, both supporting and expanding our previous findings (Rodriguez et al, 2020) showing defective NAA-induced reprogramming in *atg* mutants.

## Autophagy facilitates LR formation

Given that we have seen that autophagy impairment leads to abrogation of ARF7 oscillation and affected PBS establishment, we wondered if accumulation of ARF7 in *atg* mutants could have an impact in other ARF7-dependent phenotypes. Besides PBS, ARF7 regulatory role during LR initiation is also well established (e.g., Okushima et al, 2007; Goh et al, 2019); consequently, we examined LR production in *atg* mutants by transplanting 5-day-old seedlings to vertical plates and scored LRs after 9 days. As shown in Fig. 6A,B, *atg* mutants have a significantly lower LR density when compared to Col-0. To gain a deeper insight into the impact of defective autophagy-mediated ARF7 homeostasis during LRI we

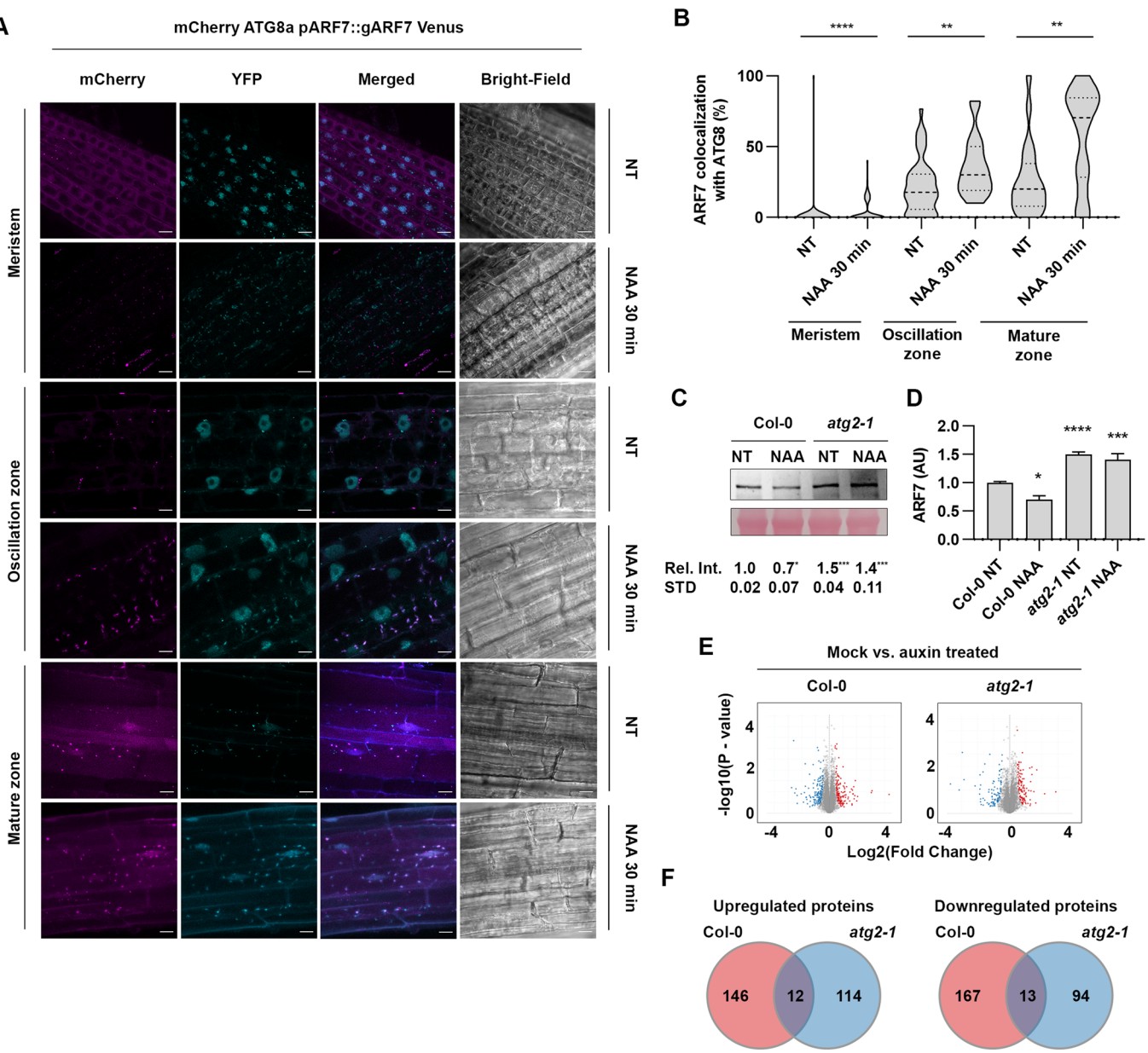

**Figure 5. NAA-induces ARF7 condensation and autophagy-mediated degradation.**

(A) Co-localization of ARF7-Venus with mCherry-ATG8a (nuclei and/or cytoplasmic foci) in meristem, oscillation or maturation zone after 30 min of treatment with 1 μM NAA. Pictures were taken in the respective root zone of 10–14-day old *Arabidopsis* seedlings. Scale bar 10 μM. (B) Quantification of the co-localization of ARF7-Venus with mCherry-ATG8a under conditions depicted in (A); The outline of the violin plots represents the probability of the kernel density. Dotted lines represent interquartile ranges (IQR), with the thick horizontal line representing the median; whiskers extend to the highest and lowest data point. Results were obtained from three independent experiments with at least eight plants per condition. Asterisks mark statistical significance to NT (MS+ solvents) according to a Mann–Whitney *U* test (**0.001; ****0.0001). ARF7 western blot (C) and quantification (D) from samples extracted from Col-0 or *atg2-1* treated with MS + solvent or MS supplemented with NAA (1 μM, 60 min). Values below each band represent the relative intensity ratio between ARF7 and RuBisCO large subunit (Rbc L) as stained with Ponceau S used as loading control. Ratios were normalized to each genotype's NT sample (MS+ solvent), which was set to 1. Values are mean ± SD of five biological replicates. Statistical analysis was done using a Holm–Sidak test (*0.05; ***0.001). (E) Volcano plot displaying pairwise protein content changes in Col-0 or *atg2-1*, 30 min after Mock (MS+ EtOH) or NAA treatment (MS + NAA) with a F.D.R. <0.05 using an unpaired *t* test. (F) Venn diagrams showing the number of differentially expressed proteins overlapping between the datasets (F.D.R. <0.05). Source data are available online for this figure.

evaluated ARF7-Venus signal in the LRPs of Col-0 and *atg2-1* seedlings. Our analysis revealed that there was significantly more ARF7 signal in *atg2-1* background than in control plants, which is consistent with our data for other root zones (Fig. 6C,D).

Importantly, we also observed that while the intensity of ARF7-Venus signal displayed oscillations over time in Col-0 background, this was not the case for *atg2-1* where the intensity of ARF7-Venus signal remained constant over the same period of time (Fig. 6C,D).

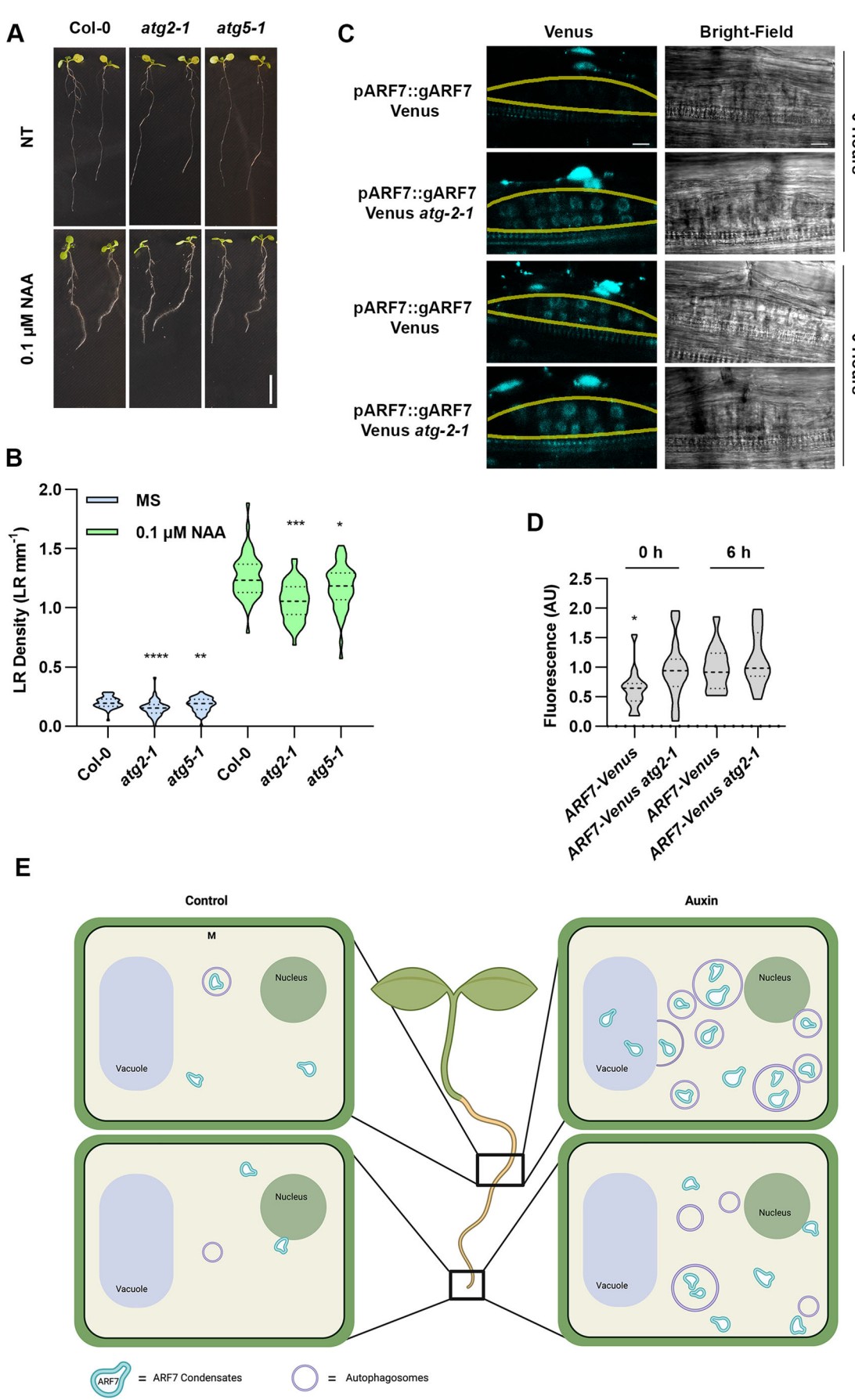

**Figure 6. Autophagy-deficient mutants display impaired proteome remodeling upon auxin treatment and show decreased LR formation capacity.**

(A, B) LR density quantification in Col-0, *atg2-1*, and *atg5-1* grown in MS plates supplemented with Solvent (MS) or 0.1 μM NAA. The outline of the violin plots represents the probability of the kernel density. Dotted lines represent interquartile ranges (IQR), with the thick horizontal line representing the median; whiskers extend to the highest and lowest data point but no more than ±1.5 times the IQR from the box. Results were obtained from three independent experiments with at least 40 plants per condition, asterisks mark statistical significance to Col-0 according to the *t* test (*0.05; **0.01; ***0.001; ****0.0001). Scale bar: 1 cm). (C, D) Confocal images of lateral root primordial in Col-0 (top) or *atg2-1* (bottom) expressing *ARF7-Venus* at given time. (D) Fluorescence intensity quantification of ARF7-Venus subcellular localization in given lines. The values presented for ARF7-Venus fluorescence intensity in *atg2-1* background were normalized to the values in Col-0 background (which was set to 1). The outline of the violin plots represents the probability of the kernel density. Dotted lines represent interquartile ranges (IQR), with the thick horizontal line representing the median; whiskers extend to the highest and lowest data point. Results were obtained from three independent experiments with at least seven plants per condition being analyzed in total. At least 14 LRPs were measured per genotype/tissue. Asterisks mark statistical significance to NT according to a Welch *t* test (*P < 0.05). Yellow line delimits the section (LRP) used for ARF7-Venus quantification (E) A hypothetical model displaying the tissue and hormone-specific regulation of ARF7. The figure was drawn using BioRender (https://www.biorender.com). Source data are available online for this figure.

Because we have seen that lack of autophagy leads to condensation of ARF7 (Fig. 3) which has been linked with decreased auxin responses (Powers et al, 2019) and because NAA treatment did not produce major changes to ARF7 abundance in *atg2-1* seedlings (Fig. 5C,D), we questioned if *atg* mutants might have impaired auxin responses. To ascertain this, we germinated Col-0 and *atg* mutants and after 5 days, transferred them to vertical MS plates with or without NAA (0.1 μM) and counted LRs 9 days after germination. As seen in Fig. 6A,B, NAA treatment caused an increase in LR density for all genotypes, but it did not rescue the differences observed between Col-0 and *atg* mutants. The same scenario was observed with *nbr1-c1*, which showed reduced LR formation when compared to Col-0 (Fig. EV5) These results agree with our previous findings (Rodriguez et al, 2020) regarding *atg* mutants reduced ability to respond effectively to callus-inducing media (rich in auxin) to form callus, an in vitro process which molecularly resembles lateral root initiation (Sugimoto et al, 2010).

## Discussion

Here, we show that autophagy modulates ARF7 and ARF19 turnover which would be mediated through the selective cargo adapter NBR1 (Figs. 1 and 2), at least for ARF7. We and others (Jing et al, 2022) have shown that MG132 treatment also causes ARF7 to accumulate, which suggests proteasomal activity also regulates this TF turnover. However, care should be taken when inferring proteasomal contribution based solely on MG132-derived data as this inhibitor produces several off-target effects like transcriptional upregulation (Wang et al, 2008; Cho et al, 2013), interference with lysosomal proteases (Garrison and Bangs, 2020) and autophagosome assembly (Ji et al, 2016). Nonetheless, as we see a synergistic effect of MG132 and autophagy inhibition in ARF7 accumulation (Fig. 1), it is likely that both degradation pathways are involved in regulating ARFs turnover. The recruitment of both degradation pathways to promote ARF7 turnover is in line with the necessity to keep ARF7 output under tight control (Moreno-Risueño et al, 2010; Perianez-Rodriguez et al, 2021) and with the fact that excessive activating ARFs lead to dramatic developmental retardation due to exaggerated auxin program execution (Powers et al, 2019). Interestingly, we find that while *atg* mutants over accumulate ARF7, autophagy blockage leads to increased ARF7 cytoplasmic condensates and increased ARF7-Venus nuclear signal. Consequently, despite having higher ARF7 levels than Col-0, *atg* mutants show diminished auxin response (Fig. 5), PBS

establishment (Fig. 4), LR formation (Fig. 6), and callus formation (Rodriguez et al, 2020), a process reminiscent of LR initiation. How does accumulation of ARFs equate to diminished auxin responses? Our data indicates that autophagy inhibition leads to ARF7 accumulation which may induce supersaturation (Mcswiggen et al, 2019) and cause ARF7 condensation in the cytoplasm (Figs. 1 and 3). Recent reports (Powers et al, 2019; Jing et al, 2022) have linked ARFs condensation with decreased auxin responsiveness, therefore, the disruption of protein degradation pathways we report here causes ARF7 levels to increase, promoting cytoplasmic aggregation and consequent reduced auxin responsiveness. ARF7 accumulation and reduced auxin responsiveness might be explained by ARF7's repressor role (Perianez-Rodriguez, 2021); while ARF7 is needed for auxin responsiveness, ARF7 activity functions in waves and an inability to terminate ARF7 activity maxima likely prevents the plant from executing appropriate auxin-induced reprogramming. While our data provides an indication that the mobilization of ARF7 to cytoplasmic foci might be part of a mechanisms aimed at adjusting ARF7 abundance and auxin responsiveness, more experiments are needed to fully understand this process. For instance, Jing et al (2022) recently showed that the loss of function of *AFF1* (F-box protein) displays reduced auxin responsiveness due to cytoplasmic hypercondensation and consequent nuclear depletion of ARFs. Despite those dramatic effects, the *aff1* mutants only display mild developmental phenotypes and surprisingly seems able to produce LRs when supplemented with 2-4D; this contrasts with *arf7 x arf19* mutant's severe LR formation defects, even when cultured with auxin (Okushima et al, 2007). Moreover, we report that ARF7 cytoplasmic foci do not co-localize with autophagosomes at the same extent in meristem vs more mature tissues, even after inducing ARF7 cytoplasmic condensation with a short NAA treatment (Fig. 5). This could be explained by the fact that meristem have an enhanced proteostatic network (Llamas et al, 2021) and thus, protein misfolding and aggregation could be less likely to occur. A caveat to this interpretation is that in young tissue, NAA-induced condensates might have greater liquid-like properties, hence the lower co-location with autophagic components. Given that condensate solidification is a progressive process, it is possible with time, ARF7 condensates in young tissue would increase solidity and increase their co-localization with ATG8a/NBR1. In sum, while we provide robust evidence indicating that ARF7 cytoplasmic condensation is part of a triage mechanism for its autophagic degradation, further clarification is needed to completely uncover how this process is regulated in different root zones.

Consistent with defective oscillation of ARF7 levels, *atg* deficiency led to decreased PBS establishment (Fig. 4) and LR formation (Fig. 6A,B). In the context of LR formation, there is evidence of autophagic participation in a wide range of processes like cell death regulation of endodermal cells above LRPs (Escamez et al, 2020) and potentially also during PIN degradation (Kleine-Vehn et al, 2008). Moreover, autophagy modulates proteome turnover induced by different hormones like auxin (Fig. 5; Rodriguez et al, 2020), cytokinin (Acheampong et al, 2020); Rodriguez et al, 2020) and ABA (Rodriguez et al, 2020); all of which are known to influence LR formation. For these reasons, deficiencies in such a transversal process like autophagy complicate result interpretation and ascribing phenotypes to a specific function of autophagy. For instance, ABA is known to interfere with LRs formation (De Smet et al, 2003) and consequently, misregulation of ABA signaling in *atg* mutants (Rodriguez et al, 2020) could be a source of defective LR formation. However, we have previously seen that *atg* mutants have difficulties forming callus (Rodriguez et al, 2020) a process resembling LR initiation and because ABA does not affect callogenesis (Sugimoto et al, 2010), it is plausible that LR deficiencies in *atg* mutants occur at an earlier stage than ABA-induced effects (LR emergence). Nonetheless, because our results unequivocally show that autophagy specifically impacts the turnover of the important LR regulators ARF7 and ARF19, it is undeniable that at the very least, defective ARFs turnover in *atg* mutants must contribute to the decreased auxin responses and LR formation we report here (Fig. 6E).

In conclusion, we have uncovered a novel role for selective autophagy during postembryonic organ development through the turnover of the key developmental regulators ARF7 and ARF19 and provided further insight into the intricate mechanisms behind ARFs fine-tuning and how this affects root architecture.

# Methods

## Plant material and phenotypic characterization

*Arabidopsis* seedlings were grown on solid, half-strength MS media (0.8% agar, 1% sucrose at 5.7 pH) and kept at 21 C° with a photoperiod of 16 h (120 µE/m$^2$/s). Seeds were sterilized with 1.3% bleach followed by 70% alcohol and then rinsed three times with sterilized water. The following lines were used throughout this study; Col-0, *atg2-1* (SALK_076727), *atg5-1* (SAIL_129_B07), *nbr1-c1* (Ji et al, 2020), *mCherry-ATG8a* & *mCherry-NBR1* (Svenning et al, 2011), pARF7::gARF7-Venus (Orosa-Puente et al, 2018), *pARF19::ARF19-GFP/nph4-1 arf19-1* (Okushima et al, 2007); *DR5::Luciferase* (Moreno-Risueno et al, 2010).

For the time course experiments, seedlings were acclimated in liquid half-strength MS media for at least 2 days before sampling. The sampling was done just before the chamber light was turned on, and then at indicated times. For LR scoring, seedlings were germinated on half-strength MS media and set to grow vertically for the indicated time. Per genotype, at least 70 plants were analyzed and derived from six independent experiments. LR's were counted at day 9 as described by Dubrovsky and Forde, 2012, and only LR's that had at least reached stage IV were included. Per genotype/condition at least 40 plants were counted derived six independent experiment. All genotypes were blinded during counting in both assays to prevent potential bias.

## Chemical treatments

For confocal microscopy and western blot assays, seedlings were acclimated in liquid half-strength MS media for 2 days acclimation before imaging. For hormone treatment effect on the co-localization of ARF7 with ATG8 & NBR1, samples were either treated with NAA (1 µM), Kinetin (1 µM), or MS for the given time. Chemical inhibition of autophagy was done by pre-treating seedlings with the vacuolar protease inhibitors E-64d (1:3000, stock 10 µM) and Pepstatin A (1 µM) 16 h prior to imaging or western blotting. For inhibition of condensates in the mature regions of the root, roots were imaged just before and directly after treatment with condensate inhibitor 1,6-hexanediol (10%) as described by (Wang et al, 2021). Images were taken at 5 min interval for 30 min. For PBS determination, seedlings were directly germinated and grown in MS plates supplemented with treatments: Mock, 1 µM Pepstatin A or 10 µM E-64d and grown vertically for 7 days before being analyzed.

For protein analysis of ARF7 in *atg* mutants upon proteasomal blockage, samples were treated with or without MG132 (100 µM) for 16 h before harvest.

For luciferase activity imaging, the plates with 7dpi seedlings were sprayed with 1.5 ml of luciferin (D-luciferin potassium salt, Biosynth Carbosynth, Ref: FL08608) 2.5 mM and let to dry. The CCD camera NightOwl and its software IndiGO were used to visualize the reporter *DR5::Luciferase* in the different treatments. We took a luminescence image with 5 min of exposition. The images were processed in the software Metamorph Microscopy Automation Software. The same software was used to measure PBS number and Oscillation Zone intensity.

## Confocal imaging

Confocal imaging of seedlings was taken with LSM-700 Zeiss confocal microscope from the Center for Advanced Bioimaging (CAB) Denmark. Images were taken using 20× air or 63× water objectives. For analysis of ARF7-Venus signal over time, we arbitrarily set the beginning of our experiments to coincide with a timepoint where ARF7 signal was low. Images were analyzed using Zen Blue 2012 (Zeiss).

## Protein extraction

Proteins were extracted using 3xSDS buffer (30% glycerol, 3% SDS, 94 nM Tris, pH 6.8, Bromophenol blue, Complete ultra-tablets EDTA-free protease inhibitor cocktail (Roche), 50 mM DTT), then samples were centrifugated for 20 min at 4 °C 14,000 rpm and the supernatant was moved into a new tube. Samples were denatured at 95 °C for 5 min before loaded it for SDS-PAGE.

## SDS-PAGE and western blotting

Protein samples were separated in a 4–20% pre-casted gel (Bio-Rad) and transferred overnight onto a nitrocellulose membrane (GE Healthcare). Membranes were blocked for 1 h with 5% skim milk (Merck) in TBST [50 mM Tris-HCL, pH 7.5 150 nM NaCl, 0.1% Tween-20 (Sigma)] and incubated 2 h at room temperature or overnight (4 °C) with primary antibodies: anti-ARF7 (Wang et al, 2005), anti-NBR1 (AS14 2805, Agrisera; 1:5000); anti-GFP (TP401

**Table 1. Primers used in this study.**

| Cloning primers | |
|---|---|
| ARF7 forward primer | 5′-CACCATGAAAGCTCCTTCATCAA-3′ |
| ARF7 reverse primer | 5′-TCACCGGTTAAACGAAGTGGC-3′ |
| NBR1 forward primer | 5′-CACCATGGAGTCTACTGCTA-3′ |
| NBR1 reverse primer | 5′-TCAAGCCTCCTTCTCCCCT-3′ |
| TSPO forward primer | 5′-GCAGGCTCCGCGGCCATGGATTCT CAGGACATCAGA-3′ |
| TSPO reverse primer | 5′-AGCTGGGTCGGCGCGTCACGCGACTG CAAGCTtabfigTTAC-3′ |
| ATG8a forward | 5′-CACCATGATCTTTGCTTGCTTGA-3′ |
| ATG8a reverse | 5′-TCATCAAGCAACGGTAAGAGAT-3′ |
| qPCR primers | |
| ACT2 forward primer | 5′-CTTGTTCCAGCCCTCGTTTGTG-3′ |
| ACT2 reverse primer | 5′-CCTTGGAGATCCACATCTGCTG-3′ |
| ARF7 forward primer | 5′-CAAGGTCACAGTGAGCAAGTCG -3′ |
| ARF7 reverse primer | 5′-TGTGGAGCATGCATATGAGCTTGG -3′ |
| LBD16 forward primer | 5′-TCCATGATCGATGTGAAGCTGTCG-3′ |
| LBD16 reverse primer | 5′-TGTGATTGCAAGAAAGCCACCTG-3′ |

AMSBio, 1:1000); anti-ATG8a (Yoshimoto et al, 2004) Membranes were washed, followed by incubation in anti-rabbit HRP conjugate (Promega; 1:5000). Home-made chemiluminescent substrate (Mruk and Cheng, 2011) was applied before detection. Bands were quantified using ImageJ and normalized to loading controls.

## Sample preparation for proteomic analysis

Samples were prepared as described by Bastrup et al, 2022, in brief tissue extracts (100 µg) were diluted in digestion buffer (0.5% SDC in 50 mM TEAB), heat-treated for 5 min at 95 °C, and prepared by a modified filter-aided sample preparation (FASP) protocol (65). In brief, tissue extracts were transferred to 0.5 ml (tilted) spin filters (Amicon), centrifuged at 14,000× g for 15 min, and reduced and alkylated by the addition of digestion buffer containing 1:50 (v:v) tris(2-carboxyethyl)phosphine (0.5 M, Sigma) and 1:10 (v:v) 2-chloroacetamide (Sigma) for 30 min at 37 °C. Samples were digested in fresh digestion buffer containing 1 µg Typsin/LysC mix (Promega) and 0.01% ProteaseMAX (Promega) O/N at 37 °C. Digested samples were desalted using stage-tips containing styrene-divinylbenzene reversed-phase sulfonate material (SDB-RPS; 3 M).

The mesenteric artery-based library was generated using a pooled digested and stage-tipped sample from the 12-week-old SHRs and WKYs that was fractionated. A high-pH reverse-phase peptide (HpH) fractionation kit (Pierce, Thermo Scientific) was used to create the 15 fractionations.

## Data acquisition by liquid chromatography–mass spectrometry (LC-MS)

Data acquisition was performed as described by Bastrup et al, 2022, in brief peptides were separated on 50-cm columns packed with ReproSil-Pur C18-AQ 1.9-µm resin (Dr Maisch GmbH). Liquid chromatography was performed on an EASY-nLC 1200 ultra-high-pressure system coupled through a nanoelectrospray source to an Exploris 480 mass spectrometer (Thermo Fisher Scientific). Peptides were loaded in buffer A (0.1% formic acid) and separated applying a nonlinear gradient of 5–65% buffer B (0.1% formic acid, 80% acetonitrile) at a flow rate of 300 nl/min over 100 min. Spray voltage was set to 2400 V. Data acquisition switched between a full-scan (120,000 resolution, 45 ms max. injection time, AGC target 300%) and 49 DIA scans with isolation width of 13.7 $m/z$ and windows overlap of 1 $m/z$ spanning a precursor mass range of 361 to 1033 $m/z$ (15,000 resolution, 22 ms max. injection time, AGC target 1000%). The normalized collision energy was set to 27. All statistical analysis of LFQ-derived protein expression data was performed using the automated analysis pipeline of the Clinical Knowledge Graph (Santos et al, 2022).

## Vector construction and plant transformation for BiFC

To create pDEST-ATG8a-SCYNE, pDEST-NBR1-SCYCE, pDEST-ARF7-VYNE and pDEST-TSPO-SCYCE, we PCR amplified (CloneAMP HIFI PCR premix, TAKARA) the coding sequences of ATG8a, NBR1, ARF7, and TSPO flanked recognition sequence for Gateway (Thermofischer) or In-Fusion (Clontech) cloning. ATG8a, NBR1 and ARF7's PCR products were cloned into pENTR (Life Technologies) and subsequentially transferred (LR clonase, Invitrogen) into the Gateway-compatible BiFC vectors (Gehl C et al, 2009) pDEST-GW SCYNE, pDEST-GWSCYCE, pDEST-GWVYNE. TSPO PCR amplicon was directly cloned into pDEST-GWSCYCE using In-Fusion cloning. Agrobacterium strain GV3101 containing constructs were grown in Yeast-extract peptone media supplied with the respective antibiotics overnight. Cultures were spun and resuspended in 10 mM MgCl2, 10 mM MES-K and 100 µM acetosyringone to the density of 600 nm = 0.15–0.2 per construct and mixed 1:1 and left at room temperature for 2 h, followed by infiltration in 3 weeks old N. benthamiana. Pictures were taken 48 hpi using a Zeiss LSM-700 confocal microscope.

## Immunoprecipitation

Immunoprecipitation assay was described previously (Zuo et al, 2022). In brief, 1 g of 14-day-old seedlings were ground with mortar and pestle and 2 ml of IP extraction buffer (50 mM Tris-HCl pH 7.5, 150 mM NaCl, 5% glycerol 1 mM EDTA, 0.1% NP40, 10 mM DTT, a protease inhibitor (mini complete ultra, EDTA-free) was added, followed by centrifugation at 4 °C 13,000 rpm, 20 min. The supernatant was incubated with 20 µl GFP trap beads (Chromotek) rotating for 2–4-h at 4 °C. The beads were washed 3–4 times with washing buffer (20 mM Tris-HCl pH 7.5, 0.15 M NaCl, 0.1% NP40). In total, 30 µL of 3×SDS was added to the beads and heated up 5 min at 95 °C before proceeding for SDS-PAGE.

## RNA extraction

RNA was extracted as described previously (Zuo et al, 2022) using TRIzol reagent (Thermo Scientific), treated with DNase (Thermo Scientific) and was reverse transcribed into cDNA using Reverse Aid kit (Thermo Scientific) as described by the manufacturer. qPCR assays were executed on the Quantstudio 1 (Applied Biosystems)

system using SYBR Green master mix (Thermo Fisher), *ACT2* was used as internal control. All primers used for qPCR analysis are listed in Table 1. The experiment was repeated three times.

## Data availability

The mass spectrometry data from this publication has been deposited to the PRIDE archive (Perez-Riverol et al, 2022; http://www.ebi.ac.uk/pride/archive/projects/PXD042834) and given the identifier [PXD042834] The source data for the microscopy has been deposited at BioImage archive (https://www.ebi.ac.uk/biostudies/bioimages/studies/S-BIAD1025).

The source data of this paper are collected in the following database record: biostudies:S-SCDT-10_1038-S44319-024-00142-5.

## Peer review information

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

## Acknowledgements

This work was funded by a Danmarks Frie Forskningsfond grant to E.R. (DFF1-1032-00249B) and by Ministerio de Ciencia e Inovacion, Agencia Estatal de Investigación (MICIN/AEI/10.13039/501100011033) of Spain, ERDF a way of making Europe and NextGenerationEU/PRTR to MAM-R (TED2021-129530B-I00 and PID2022-140719NB-I00). We would like to thank Tom Guilfoyle (anti-ARF7 antibody), Koki Yoshimoto (anti-ATG8a antibody), Malcolm Bennett (*pARF7::gARF7-Venus*), Liwen Jiang (*NBR1-C1*) and Hidehiro Fukaki (*pARF19::ARF19-GFP nph4-1 arf19-1*) for sharing material/reagents. We would also like to thank Suksawad Vongvisuttikun for technical assistance, the 2022 classes from the courses "Plant Molecular Biology" & "Experimental Higher Model Organisms" (University of Copenhagen, Denmark) for performing the ARF19-GFP & ARF7-Venus western blots, respectively. Confocal microscopy imaging was performed using equipment from the Center for Advanced Bioimaging (CAB) Denmark, University of Copenhagen. Mass spectrometry analyses were performed by the Proteomics Research Infrastructure (PRI) at the University of Copenhagen (UCPH), supported by the Novo Nordisk Foundation (NNF) (grant agreement number NNF19SA0059305).

## Author contributions

**Elise Ebstrup**: Formal analysis; Investigation; Methodology; Writing—original draft; Writing—review and editing. **Jeppe Ansboel**: Data curation; Formal analysis; Supervision; Validation; Investigation; Visualization; Methodology; Writing—original draft; Writing—review and editing. **Ana Paez-Garcia**: Formal analysis; Validation; Investigation; Methodology. **Henry Culp**: Formal analysis; Investigation. **Jonathan Chevalier**: Investigation. **Pauline Clemmens**: Investigation; Visualization; Methodology. **Nuria S Coll**: Investigation; Methodology; Writing—review and editing. **Miguel A Moreno-Risueno**: Formal analysis; Supervision; Investigation; Methodology; Writing—review and editing. **Eleazar Rodriguez**: Conceptualization; Formal analysis; Supervision; Funding acquisition; Validation; Investigation; Visualization; Methodology; Writing—original draft; Project administration; Writing—review and editing.

Source data underlying figure panels in this paper may have individual authorship assigned. Where available, figure panel/source data authorship is listed in the following database record: biostudies:S-SCDT-10_1038-S44319-024-00142-5.

## Disclosure and competing interests statement

The authors declare no competing interests.

# Expanded View Figures

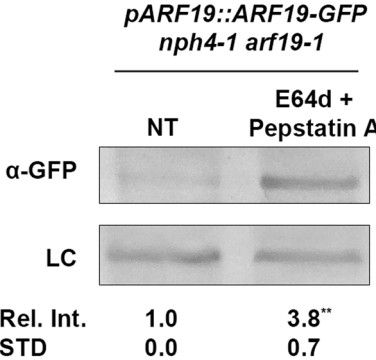

**pARF19::ARF19-GFP**
**nph4-1 arf19-1**

| | NT | E64d +<br>Pepstatin A |
|---|---|---|
| α-GFP | | |
| LC | | |
| Rel. Int. | 1.0 | 3.8** |
| STD | 0.0 | 0.7 |

**Figure EV1.  Autophagy inhibition promotes ARF19 accumulation.**

GFP western blot of 14-day-old *pARF19::ARF19-GFP/nph4-1 arf19-1* seedlings grown in MS or MS supplemented with vacuolar protease inhibitors E-64D and Pepstatin A for 24 h. Values below each band represent the ratio between ARF19-GFP and the loading control (LC) normalized to NT (MS+ solvent), which was set to 1. Image is representative of three biological replicates; asterisk depict statistical significance from NT according to a Man–Whitney *U* test (*P* ≤ 0.01). Source data are available online for this figure.

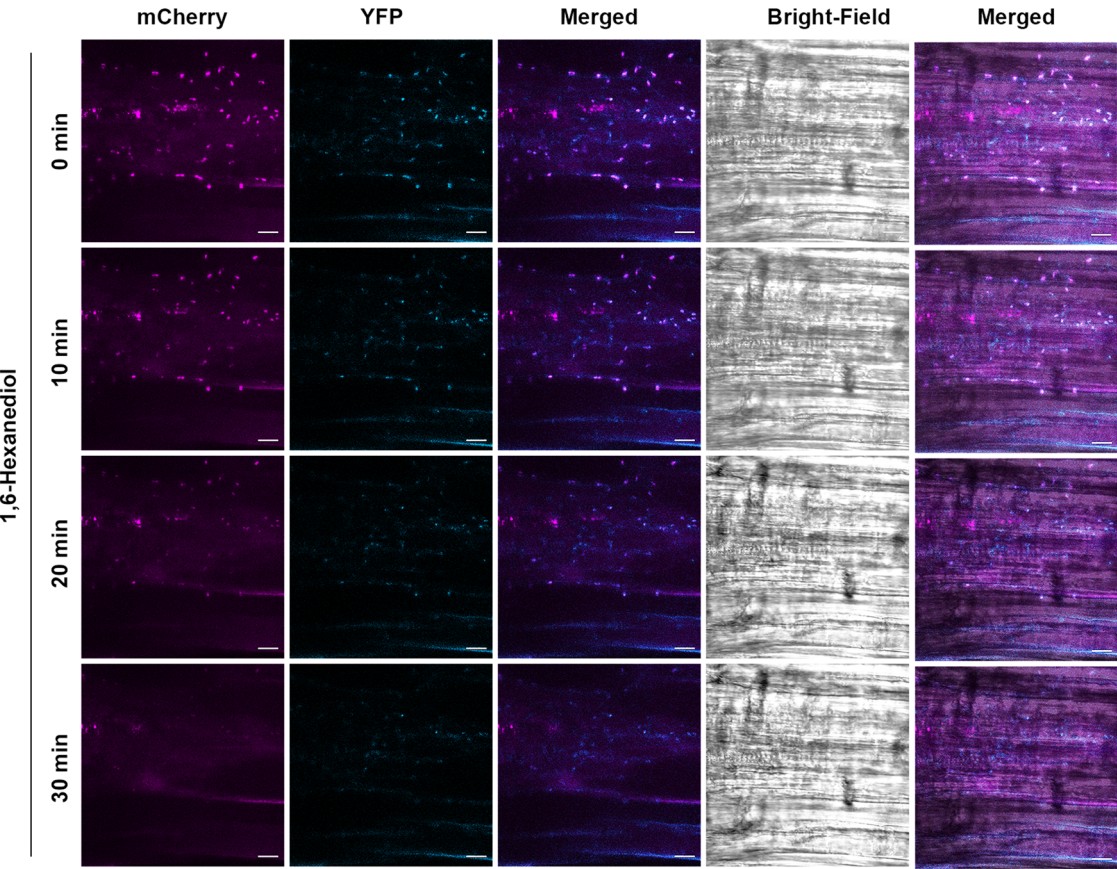

**Figure EV2. ARF7 and ATG8 condensates have liquid properties.**

*pARF7::gARF7-Venus* x *mCherry-ATG8a* plants were treated with the LLPS inhibitor 1,6-Hexanediol and imaged at given timepoints. Representative pictures were taken in the maturation zone of 10–14-day old *Arabidopsis* seedlings. Leftmost Merge column represent the merger of YFP and mCherry channels, while the rightmost also includes the T-PMT channel (brightfield). Scale bar 10 μM. Source data are available online for this figure.

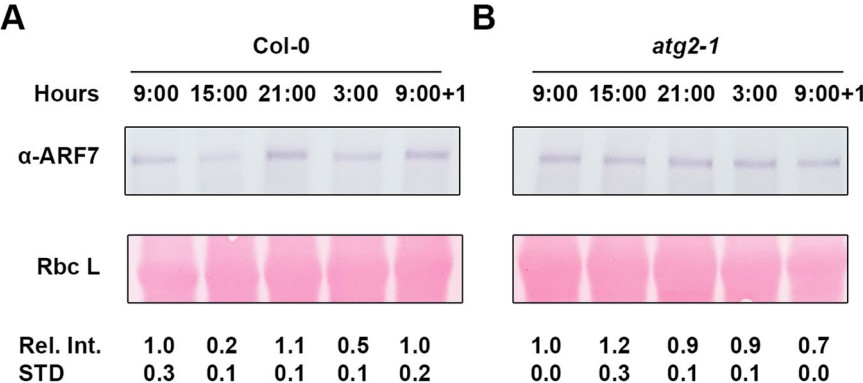

**Figure EV3. ARF7 abundance rhythmically fluctuates over 24 h and this regulation is lost in *atg2-1*.**

(A, B) ARF7 western blots from proteins extracted from Col-0 (A) and *atg2-1* (B) over a 24 h period (09:00 am, 15:00 pm, 21:00 pm, 03:00 am and 09:00 am of the following morning). Values below each band represent the relative intensity ratio between ARF7 and RuBisCO large subunit (Rbc L) as stained with Ponceau S used as loading control. Ratios were normalized to untreated (NT) which was arbitrarily set to 1. The experiment was repeated with similar results. Source data are available online for this figure.

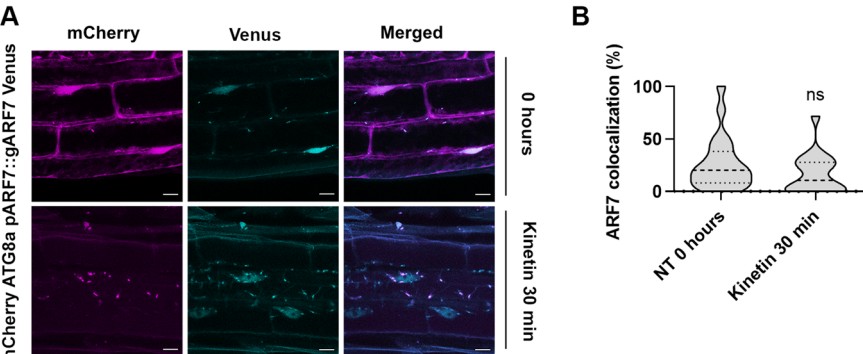

**Figure EV4. Cytokinin has no effect on ARF7 co-localization with autophagosomes.**

(A) Co-localization of ARF7-Venus with mCherry-ATG8a (nuclei and/or cytoplasmic foci) in the maturation zone. MS+ solvent (NT) or 1 µM Kinetin for 30 min were used for the treatment. Pictures were taken in the respective root zone of 10–14-day old *Arabidopsis* seedlings. Scale bar 10 µM. (B) Quantification of the co-localization of ARF7-Venus with mCherry-ATG8a. The outline of the violin plots represents the probability of the kernel density. Dotted lines represent interquartile ranges (IQR), with the thick horizontal line representing the median; whiskers extend to the highest and lowest data point. Statistical significance was done according to a Wilcoxon-rank test. At least five plants per conditions were analyzed. Source data are available online for this figure.

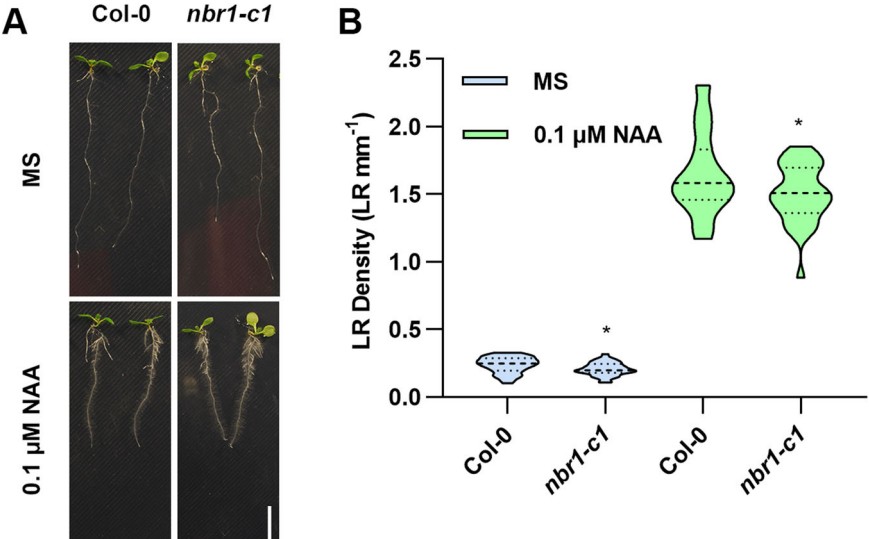

**Figure EV5.  NBR1 loss-of-function mutants show reduced LR formation.**

(A, B) LR density quantification in Col-0 and nbr1-c1 grown in MS plates supplemented with Solvent (MS) or 0.1 μM NAA. The outline of the violin plots represents the probability of the kernel density. Dotted lines represent interquartile ranges (IQR), with the thick horizontal line representing the median; whiskers extend to the highest and lowest data point but no more than ±1.5 times the IQR from the box. Results were obtained from two independent experiments with at least 25 plants per condition, asterisks mark statistical significance to Col-0 according to the *t* test (*0.05). Scale bar: 1 cm). Source data are available online for this figure.

