## [Peer Review File · EMBO Reports]

NBR1-mediated selective autophagy of ARF7 modulates root branching

Elise Ebstrup, Jeppe Ansboel, Ana Paez-Garcia, Henry Culp, Jonathan Chevalier, Pauline Clemmens, Nuria Coll, Miguel Moreno-Risueno, and Eleazar Rodriguez

Corresponding author(s): Eleazar Rodriguez (eleazar.rodriguez@bio.ku.dk)

Review Timeline:

Transfer from Review Commons:	8th Dec 23
Editorial Decision:	22nd Feb 24
Revision Received:	17th Mar 24
Editorial Decision:	3rd Apr 24
Revision Received:	5th Apr 24
Accepted:	10th Apr 24

Transaction Report: This manuscript was transferred to EMBO reports following peer review at Review Commons.

**Review
COMMONS**

Review #1

1. Evidence, reproducibility and clarity:

Evidence, reproducibility and clarity (Required)

Summary:

This manuscript shows the involvement of both the proteasome and autophagy pathways in the turnover and therefore regulation of ARF7, an auxin-responsive factor involved in lateral root formation. The authors bring crucial information for the understanding of how autophagy is involved in auxin-signaling.

Major comments:

The key conclusions appear overall convincing yet this reviewer would strongly advise to take into account the following remarks for a clearer and more convincing line of inquiry. This reviewer also believes that the additional experiments could be performed relatively fast apart for the point 9) where the establishment of a homozygous line could take 6 months or more.

1. Figure 1 & Figure EV1: The nature of the loading control should be stated as it appears to be a specific protein detected by immunoblotting. Furthermore, if the authors wish to make a stronger point as to whether ARF7 is degraded by the proteasome (considering the reserves mentioned in the Discussion section), I would recommend to perform the same assays as in Figure 1 but using an alternative proteasome inhibitor such as Bortezomib and to include a proteasome subunit KO mutant such as rpt2a-2.
2. The statement "The experiment revealed that both NBR1 (Fig 2A) and ATG8a (Fig 2B), but not free YFP, co-immunoprecipitated with ARF7-Venus." Is false as the authors did not try to co-immunoprecipitate free YFP with ARF7-Venus, they used a free YFP expressing line as a negative control for their GFP-immunoprecipitation (IP). It should further be noted that although NBR1 is detected in their free YFP IP, ATG8 is at very low levels so it should be stated that they see an enrichment of ATG8 in their ARF7-Venus IP.
3. Authors state "we were unable to detect ARF7-Venus in the input of both Co-IPs which can likely be explained by the fact that ARF7-Venus is under the control of its native promoter and thus lowly expressed.", yet putative degradation products (i.e. a smear) can be observed in the input of Figure 2A, similarly to the bands observed in both IP blots. It would be interesting to repeat these co-IPs with proteolysis inhibitors

such as MG132 or Pepstatin & E64-d to pinpoint the proteolytic machinery at the origin of ARF7-Venus degradation in the IPs.

4. Figure 2: The use of multicolor BiFC "mcBiFC" should be stated as such for an easier understanding of the reader. It would be helpful for the reader if the "GFP" signal resulting from the complementation would be highlights thanks to some arrows. Moreover, a western blot to verify the expression levels should be performed since every construct has an epitope tag as stated in Gehl et al. 2009.

5. General remark: all drug/chemical treatments performed in this study use a "non-treated" negative control, yet it should be pointed out that the correct corresponding negative controls should have the solvent used to dissolve the respective drug/chemical in order to exclude any effect of the solvent or vehicle.

6. Figure 4, Figure EV4: Considering the variability in size and staining of the Rubisco large-subunit in the 4 immunoblot panels, I would suggest blotting with another antibody such as anti-tubulin or anti-histone 3 as a loading control for a more convincing quantification. Moreover, the nature of the staining used to stain the Rubisco large-subunit should be stated. The authors also state "differences in ARF7 accumulation in atg5 compared to Col-0" yet no immunoblot is shown where both genotypes are present on the same membrane, in order to verify this statement.

7. Figure 5: In regards to LR density measurements, I recommend reading "Quantitative Analysis of Lateral Root Development: Pitfalls and How to Avoid Them" by Dubrovsky & Forde (Plant Cell, 2012) for a more robust method of evaluating lateral root density.

8. Discussion: The authors state that "autophagy blockage leads to increased ARF7 cytoplasmic condensates". To support this statement, I recommend crossing pARF7::gARF7-Venus into atg mutants and analysing the localization and the fluorescence intensity of ARF7-Venus in specific parts of the root, as well as performing immunoblotting in order to assess overall ARF7 accumulation in autophagy deficient genetic backgrounds.

****Minor comments****

9. The following statement: « In contrast, plants are able to tolerate disruption of autophagy activity without major penalties" holds true to *A. thaliana* of some other plants but it must be noted that in *O. sativa*, autophagy-deficiency may lead to male sterility, which should be considered a major penalty for evolutionary fitness. For review see Norizuki et al. 2020 (Front. Plant Sci.).

10. Figure 2: The molecular weights appear to be potentially misannotated as free YFP aligns with the 35 kDa marks although it should appear around 27 kDa.

11. Figure EV3: There are 2 merged image columns, the furthest one to the right appears to include a DIC or Trans image on top of both fluorescence channels. It would be more helpful for the reader if the DIC or Trans image was shown with the

overlay of fluorescent channels in order to assess the effect of 10% 1,6-hexandiol on the plant tissue. Moreover, demonstrating the absence of tissue damage or cell-death after 1,6-hexandiol treatment would be a plus.

12. There is a typo throughout the manuscript: ZT should be "Zeitgeber" not "zeitberg".

2. Significance:

Significance (Required)

This manuscript has the quality of describing the proteolytic balance of ARF7 and thereby, the involvement of the autophagy pathway in regulating auxin-signaling components. This research adds on to the growing interest in how autophagy participates in developmental cues, and how hormonal signaling is regulated throughout the plant.

3. How much time do you estimate the authors will need to complete the suggested revisions:

Estimated time to Complete Revisions (Required)

(Decision Recommendation)

Between 3 and 6 months

No

Review #2

1. Evidence, reproducibility and clarity:

Evidence, reproducibility and clarity (Required)

Lateral root production is a process regulated by auxin, among others. The expression of auxin-dependent genes requires the activity of transcription factors of the ARF family. In this study by Ebstrup et al., the authors suggest that selective autophagy would be involved in the degradation of the ARF7 factor involved in lateral root initiation and production in *Arabidopsis thaliana*, even though the accumulation of ARF7 in autophagy-deficiency mutant may not affect lateral root initiation.

****Major remarks and comments:****

1. In general, some experimental data do not facilitate appropriate comparisons due to lack of statistical analysis. This is particularly the case for Figures 1-a,b,c and 4-a,b,c,d.
2. Confocal microscopy images are not always convincing, due to a lack of necessary controls and also qualitatively. It would be useful, for example, to clearly indicate the objects of interest that the reader can use for comparisons. It is for example difficult to understand that chlorophyll fluorescence and GFP fluorescence (from the BiFC signal) colocalize almost in the same organelles (fig. 2c). The parent lines expressing the Venus and mCherry fusions should also serve as controls for figure EV3. Another point concerning fig. 2 a, b (IP): how do the authors explain the "GFP" signal, especially the apparent size and the doublet present only in one of the "YFP" controls after IP?
3. It would be important for the authors to clarify whether the different fluorescent fusions used are indeed functional or not. This is particularly important in the context of the proteins being studied and the possible regulatory process(es).
4. Apparently ARF7 would be degraded by the UPS system and the selective autophagy pathway. Would autophagy-deficient mutants, including *atg2-1* and *atg5-1* be more or less sensitive to MG132 (relative levels of ARF7 accumulation)? This is not clear from the data and its discussion.
5. The authors seem to insist that NBR1-mediated degradation of ARF7 by selective autophagy would be observable only preferentially in mature root tissues (probably to prevent them from forming lateral roots?). If this is the case, the title of their paper should reflect this conclusion. The authors have the tools (described in their manuscript) to unambiguously clarify this important point. Just as it would be important to demonstrate that the ARF7 proteins that accumulate would indeed be

ubiquitylated.

****Minor comments:****

1. Some of the figures would benefit from qualitative improvement, especially the photographs and micrographs.
2. The authors' attention is drawn to the existence of several typos in the text and the absence of certain references cited in the bibliography.

2. Significance:

Significance (Required)

Although the biological question is of unquestionable interest and importance, the data presented in this manuscript unfortunately do not allow us to rightly assess the contribution of this work to the state of our knowledge.

3. How much time do you estimate the authors will need to complete the suggested revisions:

Estimated time to Complete Revisions (Required)

(Decision Recommendation)

More than 6 months

Yes

Revision Plan

Manuscript number: #RC-2022-01645

Corresponding author(s): Eleazar Rodriguez

1. General Statements

We would like to thank the reviewers for noting the great relevance and potential impact of our study to the auxin, proteostasis and development communities and for their positive appreciation of our work.

Auxin is an essential phytohormone which influences a myriad of processes in plants, among these, root architecture. Central to auxin-dependent root development is the transcription factor ARF7 which controls different steps of lateral root (LR) development like definition of potential LR formation sites, LR initiation and emergence. Using a combination of phenotypic, cell biology, biochemistry, molecular biology and proteomics approaches, we provide crucial evidence of how autophagy, a cellular recycling process, is necessary for ARF7 proteostasis and the implications of autophagic mediated ARF7 turnover to root architecture. Namely, we show that:

- A) ARF7 interacts and co-localizes with autophagosome-resident protein ATG8 and selective cargo adaptor NBR1.
- B) ARF7 levels rhythmically oscillate in the whole plant and, in an autophagy-dependent manner.
- C) ARF7 proteostasis is under different spatio-temporal control, with autophagy playing a prominent role in modulating ARF7 levels in mature root tissue.
- D) Rapid auxin treatment promotes ARF7 condensation, co-localization with autophagosomes and subsequent ARF7 degradation.
- E) Accumulation of ARF7 condensates in autophagy deficient mutants correlates with abnormal auxin responsiveness and reduced LR formation in these mutants.

For these reasons, we think that our work provides novel insight and fills a crucial gap in our understanding on how ARF7 proteostasis is regulated and how this impacts auxin signaling-mediated root architecture. We also demonstrate how autophagy can directly impact plant developmental processes, through turnover of key transcription factors. Consequently, we believe our manuscript will be of high importance to researchers interested in plant development, auxin signalling and to the plant proteostasis community.

Our manuscript has been reviewed by 2 reviewers for Review Commons; their comments and our reply can be seen below. Both reviewers have stressed the importance and impactful nature of our findings and suggested more experiments to clarify/strengthen our claim, which we have followed to the best of our capacity.

Revision Plan

In the description of the planned revisions, we have done a point-by-point reply to each of the reviewer comments (our reply in **bold**)

2. Description of the planned revisions

Reviewer 1:

1) Figure 1 & Figure EV1: The nature of the loading control should be stated as it appears to be a specific protein detected by immunoblotting. Furthermore, if the authors wish to make a stronger point as to whether ARF7 is degraded by the proteasome (considering the reserves mentioned in the Discussion section), I would recommend to perform the same assays as in Figure 1 but using an alternative proteasome inhibitor such as Bortezomib and to include a proteasome subunit KO mutant such as rpt2a-2.

R: We agree with the reviewer; we have now specified the nature of the loading control in all blots. For consistency, and given reviewer 2 requests for statistics, we have now used Ponceau S-stained Rubisco L subunit as the loading control for all of our blots (also see our comment below regarding tubulin).

As for the proteasome's contribution to ARF7 degradation, we agree that a combinatorial approach should be done in order to conclude that the proteasome is the sole process mediating turnover of any given protein. Chemical inhibitors often have off target effects that seem to impact autophagic flux (Bortezomib e.g. Periyasamy-Thandavan et al 2010, Autophagy). Genetic disruption of the proteasome is equally cumbersome (e.g. rpt2a-2 still displays proteasome dependent but 26s independent protein turnover; Ueda et al 2004; Development).

In our case, however we have used MG132 also in combination with genetic disruption of autophagy. In this condition, MG132 should not further affect autophagic flux and because we see enhanced accumulation of ARF7 as compared to *atg* mutants or MG132 treated samples, we can safely conclude that both autophagy and the proteasome participate in the degradation of ARF7. Thus, there is no need to execute further experiments.

2) The statement "The experiment revealed that both NBR1 (Fig 2A) and ATG8a (Fig 2B), but not free YFP, co-immunoprecipitated with ARF7-Venus." Is false as the authors did not try to co-immunoprecipitate free YFP with ARF7-Venus, they used a free YFP expressing line as a negative control for their GFP-immunoprecipitation (IP). It should further be noted that although NBR1 is detected in their free YFP IP, ATG8 is at very low levels so it should be stated that they see an enrichment of ATG8 in their ARF7-Venus IP.

R: We thank the reviewer for this comment and apologize for this mistake. Indeed, what we meant to say is that NBR1 and ATG8 co-precipitate with ARF7-Venus but not with the GOT1-YFP line (control). Text has been corrected accordingly. We are unsure about the comment regarding NBR1 detection in the free "YFP line", as we cannot see any NBR1

Revision Plan

bands in that line. As for ATG8, we have modified the text accordingly to reflect ATG8s enrichment (L161)

3) Authors state "we were unable to detect ARF7-Venus in the input of both Co-IPs which can likely be explained by the fact that ARF7-Venus is under the control of its native promoter and thus lowly expressed.", yet putative degradation products (i.e., a smear) can be observed in the input of Figure 2A, similarly to the bands observed in both IP blots. It would be interesting to repeat these co-IPs with proteolysis inhibitors such as MG132 or Pepstatin & E64-d to pinpoint the proteolytic machinery at the origin of ARF7-Venus degradation in the IPs.

R: We performed the experiment suggested by the reviewer using either MG132 or Pepstatin & E64-d (see figure below, anti-GFP blot). We do not see major changes in these putative degradation products among samples, thus we are unsure what to conclude. We do note that the same bands can be also observed in Orosa-Puente et al 2018.

4) Figure 2: The use of multicolor BiFC "mcBiFC" should be stated as such for an easier understanding of the reader. It would be helpful for the reader if the "GFP" signal resulting from the complementation would be highlights thanks to some arrows. Moreover, a western blot to verify the expression levels should be performed since every construct has an epitope tag as stated in Gehl et al. 2009.

R: We agree with the reviewer and modified the text accordingly (L165-167). We also improved the figure to facilitate readability by adding insets and arrows. Regarding the WB of the BiFC constructs, we tried performing the westerns requested but we were unable to detect any specific bands (even at high exposure) for any of the tags (see

Revision Plan

figure below). Moreover, through personal communication, Prof. Morten Petersen's (U. Copenhagen), has told us they have also tried and failed to detect these tags by western blot. So, it could be possible that those epitopes are simply not detectable. In this context, the original article reporting these mcBiFC (Gehl et al., 2009) did not perform any confirmatory western blots to check the expression levels of their constructs. Additionally, this confirmation of expression levels was not reported for several publications using the same mcBiFC system (e.g. Licasui et al 2011, Nature; Soundappan et al 2015, The Plant Cell; Charpentier et al 2016, Science; Wang et al 2016, Nature; Zhang et al 2013, The Plant Cell; Acharya et al 2013, New Phytologist; Zhou et al 2019; Cell). At any rate, while we agree that it would be ideal to have those controls, we do not think that their absence impairs the validity of our findings as they are supported by other methods.

5) General remark: all drug/chemical treatments performed in this study use a "non-treated" negative control, yet it should be pointed out that the correct corresponding negative controls

Revision Plan

should have the solvent used to dissolve the respective drug/chemical in order to exclude any effect of the solvent or vehicle.

R: We agree with the reviewer and we did actually used MS + solvent in our “non-treated samples”, this was supposed to have been stated with a sentence in the methods but it got lost between versions; we apologize for this. We have now clarified this issue by stating so in each figure legend, which should also be better than having it just in the methods.

6) Figure 4, Figure EV4: Considering the variability in size and staining of the Rubisco large-subunit in the 4 immunoblot panels, I would suggest blotting with another antibody such as anti-tubulin or anti-histone 3 as a loading control for a more convincing quantification. Moreover, the nature of the staining used to stain the Rubisco large-subunit should be stated. The authors also state "differences in ARF7 accumulation in *atg5* compared to Col-0" yet no immunoblot is shown where both genotypes are present on the same membrane, in order to verify this statement.

R: We apologize for not stating clearly that we used Ponceau S staining to label RBSC L subunit. We note that RBSC L (Ponceau S staining) is one of, if not the most common loading controls used for WB in plant studies and that ImageJ density-based quantification of the bands presented should account for differences in band dimensions and intensities. Still, we tried repeating these blots with tubulin (AS10 680 agrisera) but we noticed (figure below) that there is a big discrepancy between the levels of Tubulin per sample and the apparent levels of total protein (as seen with whole-membrane Ponceau S staining); in some instances they are opposite. Given that Ponceau staining better reflects the amount of total protein loaded into each well, our interpretation here is that tubulin is not suitable as a loading control for these conditions. Additionally, the values we are presenting for the ratios of ARF7 and loading control are now based on averages \pm SD of at least 3 biological independent experiments and we will also provide those blots in the final version of the submission. We are confident that this approach should be sufficient to dissipate any doubts regarding the western blots. Regarding figure EV4 (now EV3) it was counterintuitive to see that short NAA treatment induced ARF7 accumulation, especially because we see enhanced co-localization with ATG8a for the same timepoint. We reasoned that this was due to kinetics of the process, i.e., some time would be needed to detect ARF7 turnover, and this was argued in the discussion before. However, we have now tested if 1h NAA treatment would be enough to see ARF7 degradation and indeed our new Figure EV3 shows that this treatment causes significant decrease in ARF7 levels. Because this simplifies our data, we have updated our figure and text to reflect these novel blots.

We cannot find in the sentence “differences in ARF7 accumulation in *atg5* compared to Col-0” stated by the reviewer; however, it is incorrect that there are no immunoblots in which *atg5-1* and Col-0 are present in the same membrane, the immunoblot in the previous figure 1B precisely has samples from both genotypes. Moreover, our new figure 1A has Col-0 *atg2-1*, *atg5-1* and *nbr1c* in the same blot, and it again shows difference in ARF7 among the genotypes.

Revision Plan

7) Figure 5: In regards to LR density measurements, I recommend reading "Quantitative Analysis of Lateral Root Development: Pitfalls and How to Avoid Them" by Dubrovsky & Forde (Plant Cell, 2012) for a more robust method of evaluating lateral root density.

R: We thank the reviewer for pointing this out. We have now addressed this issue to better present our LR density measurements and cited the paper accordingly.

8) Discussion: The authors state that "autophagy blockage leads to increased ARF7 cytoplasmic condensates". To support this statement, I recommend crossing *pARF7::gARF7-Venus* into *atg* mutants and analysing the localization and the fluorescence intensity of ARF7-Venus in specific parts of the root, as well as performing immunoblotting in order to assess overall ARF7 accumulation in autophagy deficient genetic backgrounds.

R: We agree with the reviewer and precisely because of that, we performed the cross between *pARF7::gARF7-Venus* to *atg2-1*. As it can be seen in new figure 1, we have added a western blot and confocal microscopy with quantification of GFP in nuclei vs cytoplasm comparing ARF7-venus with *atg2-1* x ARF7-venus in different root zones. These novel data corroborate our initial observations indicating that ARF7 accumulates in autophagy deficient mutants.

Minor comments

9) The following statement: ❖ In contrast, plants are able to tolerate disruption of autophagy activity without major penalties" holds true to *A. thaliana* of some other plants but it must be noted that in *O. sativa*, autophagy-deficiency may lead to male sterility, which should be considered a major penalty for evolutionary fitness. For review see Norizuki et al. 2020 (Front. Plant Sci.).

R: This is correct, as a matter of fact, in *Arabidopsis* male autophagy is also important

Revision Plan

for male fertility (<https://doi.org/10.1101/2022.08.07.503073>). Moreover, we have also noticed that *atg* mutants produce less seeds and the seeds display decreased longevity when compared to wild-type plants. While we do agree that these are penalties for evolutionary fitness, we argue that contrary to what is seen in animals, defective autophagy in plants does not normally cause critical developmental abnormalities or lethality.

10) Figure 2: The molecular weights appear to be potentially misannotated as free YFP aligns with the 35 kDa marks although it should appear around 27 kDa.

R: We would like to thank both reviewers for their comments regarding the annotation of the YFP control line used for the IP, we have realized our annotation causes confusion that requires clarification. The line used as negative control is part of the WAVE collection generated by Nico Geldner (Geldner et al 2009, Plant Journal). Here we have used WAVE LINE 18, which is a YFP fusion to GOT1 (At3g03180). GOT1 is a 21kDa protein involved in vesicle between Golgi and ER. YFP-GOT1 fusion should have 45 kDa and this fits with the annotation in the western blot. This also explains the doublet seen in the blot pointed by reviewer 2. We deeply apologize for this mistake; calling the line just “YFP-WAVE” or “YFP control line” is incomplete and causes unnecessary confusion. We have now corrected the description of the line in the figure, manuscript text and also cited the reference for this line.

11) Figure EV3: There are 2 merged image columns, the furthest one to the right appears to include a DIC or Trans image on top of both fluorescence channels. It would be more helpful for the reader if the DIC or Trans image was shown with the overlay of fluorescent channels in order to assess the effect of 10% 1,6-hexandiol on the plant tissue. Moreover, demonstrating the absence of tissue damage or cell-death after 1,6-hexandiol treatment would be a plus.

R: We are unsure of what the reviewer means by “Trans image on top of both fluorescence channels”. The image provided is a merged version of all the channels in which we decreased the input of the “trans” in the final figure to facilitate observation of the fluorescence. We have edited the figure to include the single Trans image so that tissue integrity can be appreciated but we also notice that the time frame and concentration of 1,6-hexandiol used is in line with other reports in plants and do not normally cause tissue damage.

12) There is a typo throughout the manuscript: ZT should be "Zeitgeber" not "zeitberg".

R: We thank the reviewer for pointing out this mistake and we have corrected the text accordingly.

Reviewer #1 (Significance (Required)):

This manuscript has the quality of describing the proteolytic balance of ARF7 and thereby, the involvement of the autophagy pathway in regulating auxin-signaling components. This research adds on to the growing interest in how autophagy participates in developmental cues, and how hormonal signaling is regulated throughout the plant.

R: We thank the reviewer for the positive appreciation.

Reviewer #2 (Evidence, reproducibility and clarity (Required)):

Revision Plan

Lateral root production is a process regulated by auxin, among others. The expression of auxin-dependent genes requires the activity of transcription factors of the ARF family. In this study by Ebstrup et al., the authors suggest that selective autophagy would be involved in the degradation of the ARF7 factor involved in lateral root initiation and production in *Arabidopsis thaliana*, even though the accumulation of ARF7 in autophagy-deficiency mutant may not affect lateral root initiation.

Major remarks and comments:

1. In general, some experimental data do not facilitate appropriate comparisons due to lack of statistical analysis. This is particularly the case for Figures 1-a,b,c and 4-a,b,c,d.

R: We understand and agree with the reviewer's concern, which stems from data reproducibility. We have now performed statistics for all figures requested by the reviewer and our claims are validated by these analyses. We believe a more suitable alternative to statistics is to present several blots for appreciation. Consequently, for every WB figure, we provide, as supplements, all the blots derived from biological independent experiments which should ensure the robustness of our observations.

2. Confocal microscopy images are not always convincing, due to a lack of necessary controls and also qualitatively. It would be useful, for example, to clearly indicate the objects of interest that the reader can use for comparisons. It is for example difficult to understand that chlorophyll fluorescence and GFP fluorescence (from the BiFC signal) co-localize almost in the same organelles (fig. 2c). The parent lines expressing the Venus and mCherry fusions should also serve as controls for figure EV3. Another point concerning fig. 2 a, b (IP): how do the authors explain the "GFP" signal, especially the apparent size and the doublet present only in one of the "YFP" controls after IP?

R: We agree that the figures were of lower quality, and this is due to the fact that we only had the figures included in the PDF file and not as separate high-quality versions, for this we apologize. Regarding controls for figure EV3 (now EV2) we did not consider the parental lines as necessary controls because it is known that both ARF7 and ATG8/LC3 undergo liquid-liquid phase separation and we were just confirming that the cytoplasmic structures in which there is co-localization between ARF7 and ATG8 are sensitive (at least in part) to 1,6 Hexanediol. Nonetheless, we have now performed the experiment with the parental lines (ARF7-Venus left and ATG8a right) and provide these for reviewer appreciation.

The co-localization of GFP and Chlorophyll is due to chloroplast autofluorescence rather than GFP reconstitution. We agree and thank the reviewer for suggesting the usage of arrows to indicate reconstitution of GFP; which would have avoided this situation and will improve clarity. We have now edited the figure and even provide an inset to facilitate appreciation of the BiFC reconstitution.

For the IP, we thank the reviewer for the comment, as together with reviewer 1's comment (#10), helped us realize we needed to clarify the nature of the WAVE line used

Revision Plan

as control. Please see the response to reviewer 1's comment #10 for a detailed response regarding this issue.

3. It would be important for the authors to clarify whether the different fluorescent fusions used are indeed functional or not. This is particularly important in the context of the proteins being studied and the possible regulatory process(es).

R: We understand this request by the reviewer, however these fluorescent lines (ARF7-Venus, mCherry-ATG8 and mCherry-NBR1) were not created by us and have been characterized regarding their functionality and subsequently used in many publications (ARF7 venus: Orosa-Puente et al 2018; Powers 2019, Jing et al 2022; mCherryATG8/mCherryNBR1: Suttangkakul et al 2011; Svenning et al., 2011; Li et al 2014). We thus think that this issue was solved in those publications and do not warrant follow up from our side.

4. Apparently ARF7 would be degraded by the UPS system and the selective autophagy pathway. Would autophagy-deficient mutants, including *atg2-1* and *atg5-1* be more or less sensitive to MG132 (relative levels of ARF7 accumulation)? This is not clear from the data and its discussion.

R: That is indeed an interesting question and it had already been addressed in the previous version of figure 1A-C, were it was evident that simultaneous usage of MG132 and genetic disruptions of autophagy caused further increase in ARF7 levels when compared to single conditions. This result was described in Lines 108-112 and discussed in Lines 344-352. We have now changed figure 1A to include Col-0 and all *atg* mutants tested in the same blot with or without mg132 treatment; again it can be seen that blocking both degradation pathways further increases ARF7 levels in comparison to inhibiting only one of the pathways.

Revision Plan

5. The authors seem to insist that NBR1-mediated degradation of ARF7 by selective autophagy would be observable only preferentially in mature root tissues (probably to prevent them from forming lateral roots?). If this is the case, the title of their paper should reflect this conclusion. The authors have the tools (described in their manuscript) to unambiguously clarify this important point. Just as it would be important to demonstrate that the ARF7 proteins that accumulate would indeed be ubiquitylated.

R: We appreciate the nature of the comment and agree it would be interesting to evaluate if NBR1 is mainly co-localizing with ARF7 in the mature tissue. The reason we did not do this initially is because other cargo adaptors (e.g. DSK2 which has been shown to bind ubiquitinated cargo and functions as a cargo adaptor for autophagy- Nolan et al 2017, Dev. Cell) could also participate in ARF7 degradation. Therefore we used ATG8 instead to circumvent that possibility.

Regardless, we have now analyzed NBR1's co-localization with ARF7 in different root zones. Like, for ATG8, NBR1 and ARF7 co-localize in cytoplasmic foci, especially in the maturation zone. However, there are still instances of co-localization of ARF7 with both autophagic markers in the root tip and as seen with figure 3, co-localization in the root tip is also enhanced (albeit at lower levels than in the maturation zone) by NAA treatment. We are unsure if changing the title would be restrictive, but we have produced a new title and we leave it at the reviewer/editors' discretion to indicate what they would prefer.

Regarding the ubiquitination of ARF7, it has been recently published (Jing et al 2022, Nature Comm.) that ARF7 interacts with the E3 ubiquitin ligase AFF1 and that AFF1 KOs accumulate ARF7. Moreover, ARF7 is also degraded via the proteasome (Fig 1) which requires (with few exceptions) ubiquitylation of targets. All together, these data heavily imply that ARF7 is ubiquitylated (and supports NBR1-ARF7 autophagic triage).

We have performed an IP of ARF7 Venus (see IP in response to question 3 from reviewer 1) and probed with anti-ubiquitin antibody (Agrisera AS08307) which shows that either autophagy or proteasomal inhibition leads to an increase in ARF7 levels (band, high MW smearing and even the detection of a higher MW band with a size consistent with ARF7 dimers), all of which are consistent with ARF7 ubiquitination.

Minor comments:

1. Some of the figures would benefit from qualitative improvement, especially the photographs and micrographs.

R: We apologize for the low quality of the figures; this is surely due to the compression of the images during the creation of the PDF file. We have now uploaded separate, high-resolution version of the figures to prevent this issue.

2. The authors' attention is drawn to the existence of several typos in the text and the absence of certain references cited in the bibliography.

R: We apologize for these mistakes and have corrected the manuscript accordingly.

Reviewer #2 (Significance (Required)):

Although the biological question is of unquestionable interest and importance, the data presented in this manuscript unfortunately do not allow us to rightly assess the contribution of this work to the state of our knowledge.

Revision Plan

We thank the reviewer for the positive appreciation of our manuscript, and we hope that the modifications we will perform will satisfy these remarks.

3. Description of the revisions that have already been incorporated in the transferred manuscript

4. Description of analyses that authors prefer not to carry out

- We will not perform additional experiments to address the proteasomal contribution to ARF7 degradation (either through genetic or chemical disruption of proteasomal activity). We believe that our explanation in text about MG132 off target effects in autophagic process is directed towards inferring the role of the proteasome as the sole responsible for ARF7 turnover. Simultaneous disruption of both autophagy and MG132 treatment leads to further increase in ARF7 which indicates that both pathways are involved in ARF7 turnover.

Dear Dr. Rodriguez

Thank you for the submission of your revised manuscript to EMBO reports. Please note that your manuscript was reviewed from the editorial side before transfer to EMBO reports. We have sent it back to the 2 referees from Review Commons asking them to evaluate whether their concerns have been adequately addressed. We have meanwhile received the reports from both referees that are copied below. I apologize for the delay in handling your manuscript, but I have discussed the reports further with referee #1 and with the editorial team.

As you will see, referee #2 is very positive about the study and supports publication.

Referee #1 is concerned about the lack of evidence for a physiological role of NBR1-mediated selective autophagy in ARF7 turnover, root branching/LR formation. Upon further discussion of this point, the referee suggested analysing whether the lack of NBR1 alters auxin-related phenotypes by growing seedlings and monitoring LR formation or by performing auxin treatment on *nbr1* mutant plants. We note this issue was raised for the first time. In our assessment this is in fact a reasonable request as the experiment is defined and doable. However, as the issue was not raised in the original review, we encourage it, but will not require it. However, we do agree with the referee that the conclusions have to be slightly adapted and toned down in the absence of experimental evidence that NBR1-induced autophagy affects auxin-related phenotypes, since autophagy mutants might indeed have more pleiotropic effects.

The first two points on Figure 1 and 1C relate to control experiments and should be addressed experimentally. The general comment on the relative contribution of autophagy and AFF1 on ARF7 degradation and a mutational analysis of the AIMs in ARF7 were raised for the first time as well, and while both requests are very reasonable in our opinion and the experiments are well defined and doable, we again encourage but will not require these. Please address these points, in particular the potentially synergistic or redundant roles of AFF1 and autophagy in the manuscript.

Concerns regarding Figure 4F: the suggestion to repeat experiments in the *nbr1* mutant can be addressed experimentally but we do not require transcriptomics experiments. Please discuss all other points in a point-by-point response and in the manuscript, as applicable.

I am also happy to discuss the revision further via e-mail or a video call, if you wish.

Please format your manuscript according to the guidelines below before resubmission. We will require:

2) individual production quality figure files as .eps, .tif, .jpg (one file per figure).

Please download our Figure Preparation Guidelines (figure preparation pdf) from our Author Guidelines pages <https://www.embopress.org/page/journal/14693178/authorguide> for more info on how to prepare your figures.

4) a complete author checklist, which you can download from our author guidelines (). Please insert information in the checklist that is also reflected in the manuscript. The completed author checklist will also be part of the RPF.

5) Please note that all corresponding authors are required to supply an ORCID ID for their name upon submission of a revised manuscript (). Please find instructions on how to link your ORCID ID to your account in our manuscript tracking system in our Author guidelines

()

6) We replaced Supplementary Information with Expanded View (EV) Figures and Tables that are collapsible/expandable online. A maximum of 5 EV Figures can be typeset. EV Figures should be cited as 'Figure EV1, Figure EV2' etc... in the text and their respective legends should be included in the main text after the legends of regular figures.

- Additional Tables/Datasets should be labeled and referred to as Table EV1, Dataset EV1, etc. Legends have to be provided in

a separate tab in case of .xls files. Alternatively, the legend can be supplied as a separate text file (README) and zipped together with the Table/Dataset file.

7) Before submitting your revision, primary datasets (and computer code, where appropriate) produced in this study need to be deposited in an appropriate public database (see < <https://www.embopress.org/page/journal/14693178/authorguide#dataavailability>>).

The accession numbers and database should be listed in a formal "Data Availability " section (placed after Materials & Method) that follows the model below (see also < <https://www.embopress.org/page/journal/14693178/authorguide#dataavailability>>). Please note that the Data Availability Section is restricted to new primary data that are part of this study.

Data availability

Additional information on source data and instruction on how to label the files are available .

10) Figure legends and data quantification:

- the name of the statistical test used to generate error bars and P values,
 - the number (n) of independent experiments (please specify technical or biological replicates) underlying each data point,
 - the nature of the bars and error bars (s.d., s.e.m.)
- If the data are obtained from n {less than or equal to} 5, show the individual data points in addition to the SD or SEM.
 - If the data are obtained from n {less than or equal to} 2, use scatter blots showing the individual data points.

11) Our journal encourages inclusion of *data citations in the reference list* to directly cite datasets that were re-used and obtained from public databases. Data citations in the article text are distinct from normal bibliographical citations and should directly link to the database records from which the data can be accessed. In the main text, data citations are formatted as follows: "Data ref: Smith et al, 2001" or "Data ref: NCBI Sequence Read Archive PRJNA342805, 2017". In the Reference list, data citations must be labeled with "[DATASET]". A data reference must provide the database name, accession number/identifiers and a resolvable link to the landing page from which the data can be accessed at the end of the reference. Further instructions are available at .

12) As part of the EMBO publication's Transparent Editorial Process, EMBO Reports publishes online a Review Process File to accompany accepted manuscripts. This File will be published in conjunction with your paper and will include the referee reports, your point-by-point response and all pertinent correspondence relating to the manuscript.

You are able to opt out of this by letting the editorial office know (emboreports@embo.org). If you do opt out, the Review Process File link will point to the following statement: "No Review Process File is available with this article, as the authors have

chosen not to make the review process public in this case."

Kind regards,

Referee #1:

In the present manuscript the authors try to investigate how autophagy, in particular NBR-selective autophagy is regulating root branching, by targeting transcription factor ARF7. Although, the authors provide some additional experiments, they do not present any physiological evidence that NBR1-mediated selective autophagy is indeed required for ARF7 turnover. Thus, I think the title, abstract and also claims are not appropriate and require many additional experiments to provide evidence for the contribution of selective autophagy in root branching.

Major points:

Figure 1: To me the presented blot is not convincing since RbcL is more abundant in all samples except Col-0 NT. It is also not an adequate loading control as it is an autophagy target itself. The authors argued that tubulin blots do not reflect protein loading, which I have never observed so far. If this is a major issue in their hands other antibodies such as histone, actin etc. could be used. However, I strongly recommend using a "stain-free" system or something similar to quantify protein amounts since this is an essential point in their manuscript. I would also include a fractionation experiment to see whether ARF7 accumulates in different compartments (nucleus vs. cytoplasm).

Figure 1C: Since the authors claim NBR1 is degrading ARF7, it is essential to repeat this blot in the NBR1 mutant background.

General comment: As F box protein AFF1 regulates ARF7 accumulation and nucleo-cytoplasmic partitioning it is still not clear to me why autophagy would need to regulate protein abundance of ARF7- is it only for the cytoplasmic pool? However, the image analysis shows that the nuclear signal is also increased. At this point I suggest performing fractionation experiments to reveal whether protein levels of ARF7 are altered in different compartment comparing atg mutants (and nbr1) with Col-0. I also suggest monitoring whether ARF7 is still degraded by selective autophagy in aff1 mutant background. It could be also possible that AFF1 plays a role in its autophagic degradation. Considering previous findings on AFF1-ARF7 it is essential to include a more detailed analysis to decipher the fate of ARF7.

Figure 2: Since the authors state that ARF7 harbours AIMs, I strongly recommend mutating them and repeating interaction assays with ATG8 to strengthen their findings. AlphaFold-Multimer prediction might be very helpful for this. For what does ARF7 need NBR1, if ARF7 has functional AIMs and can interact with ATG8?

Did the authors test if ATG8 or NBR1 are present in the IPs in A and B? It would be more convincing and also consistent with their mcBiFC in 2C.

Since NBR1 binds to ubiquitinated proteins, it would be important to show that ubiquitinated ARF7 is interacting with NBR1. This could be done e.g. in the presence of MG132 (blot in rebuttal letter). Additionally, to understand if AFF1 plays a role in ARF7 degradation by autophagy I suggest performing these experiments in the aff1 mutant background.

Comment to mcBiFC: The authors show that the interaction of NBR1-ATG8-ARF7 also occurs in the nucleus. How is ARF7 then "extracted" from the nucleus and degraded by autophagy? It appears to be rather complicated...Or is this an artefact of the mcBiFC method.

I also strongly recommend strengthening the findings by in vitro assays with NBR1 (K11 mutant)-ARF7 and ATG8.

Additionally, I would repeat all interaction data in the presence or absence of IAA (in the light of Figure 5) and also perform fractionations prior IPs to elucidate if the interaction is in the nucleus or cytoplasm.

Figure 4F: What about the expression of LBD16/33 as well as other auxin responsive genes in the *nbr1* mutant? If the authors want to provide evidence that NBR1 is degrading ARF7 they need to repeat this in the *nbr1* mutant. Instead of the proteomics in Figure 5, I would strongly suggest performing a transcriptomic experiment in *atg2* and *nbr1* (-/+IAA) as ARF7 is a transcription factor. If autophagy is essential for ARF7 turnover I would expect some changes in auxin responsiveness in the mutants. Otherwise, I would suspect that the phenotypes presented in Figure 6 are due to the pleiotropic nature of *atg* mutants (see comments below).

Figure 4G: To me it is not clear why this experiment was not performed in *atg* mutants. Pepstatin is inhibiting aspartyl proteases, and this assay only demonstrates that they seem to play a role in this. It is essential to perform this experiment in autophagy deficient mutant *atg2* as well as in *nbr1*.

I cannot find any description of Figure 4J-L in the results section.

Figure 5: NAA also has other effects, e.g. on endocytosis and the endomembrane system (Narasimhan et al., 2021, Plant Phys). Thus, the authors should repeat their experiments using IAA (Figure 5A-C).

5E/F to me it is not really clear what is up-/downregulated - do the authors only look into auxin responsive genes? Since ARF17 is a transcription factor it is more essential to look into the rapid transcriptional response (see comment above).

Figure 6: The authors claim in the title and abstract that ARF7 is degraded by NBR1 selective autophagy but do not show any physiological data in Figure 6: How does NBR1 respond to auxin? What about LR formation in *nbr1* and other auxin-related phenotypes? It is not surprising and known that autophagy deficient mutants have developmental defects. Thus, it is more important that the *nbr1* mutant displays these phenotypes as autophagy mutants are more pleiotropic.

Referee #2:

The authors have satisfactorily addressed my comments and concerns in their revised manuscript, resulting in notable improvements to the document.

Referee #1:

In the present manuscript the authors try to investigate how autophagy, in particular NBR-selective autophagy is regulating root branching, by targeting transcription factor ARF7. Although, the authors provide some additional experiments, they do not present any physiological evidence that NBR1-mediated selective autophagy is indeed required for ARF7 turnover. Thus, I think the title, abstract and also claims are not appropriate and require many additional experiments to provide evidence for the contribution of selective autophagy in root branching.

R: We clarify that in the first version of this manuscript, the focus of the story was on the role of autophagy in ARF7 turnover, but not specifically about NBR1. While we explained in that rebuttal why we did not want to overly highlight NBR1's role in ARF7 degradation, we changed our title, abstract and performed several additional experiments to accommodate reviewer 2's requests. We point that reviewer 2 is completely satisfied. In contraposition, reviewer 1 made no comments regarding NBR1 in that instance, and given EMBO's praxis on single review rounds, we find the nature of the new comments by this reviewer perplexing.

Nonetheless, from the battery of newly requested experiments by reviewer 1, we understood that checking for "physiological evidence" of NBR1 mediated selective autophagy could be done in reasonable time and could provide interesting information directly related to our core message. We conducted those experiments and we can demonstrate that *nbr1-c1* mutants produce significantly less LRs than *col-0* and this difference is not rescued by NAA supplementation (new figure EV5). This should rebuff any claims of pleiotropism or uncertainty regarding NBR1-mediated selective autophagy of ARF7.

As a side note, we would like to point at the nitid contrast between the first and second round of reviews by reviewer 1. In the first round, this reviewer made judicious suggestions to strengthen our claims without asking to pursue novel direction, even showing mindfulness in not demanding long-winded experiments (e.g. crosses). Now most of the requests are either: a) argumentative and have virtually no-added value to strengthen our claims, b) long-winded (+6 month) and derivative to the core message of the story (e.g. insistence in performing crosses to AFF1 or extensive protein-domain interaction studies). Below, we will argue point by point, why we understand this to be so.

Major points:

Figure 1: To me the presented blot is not convincing since RbcL is more abundant in all samples except Col-0 NT. It is also not an adequate loading control as it is an autophagy target itself. The authors argued that tubulin blots do not reflect protein loading, which I have never observed so far. If this is a major issue in their hands other antibodies such as histone, actin etc. could be used. However, I strongly recommend using a "stain-free" system or something similar to quantify protein amounts since this is an essential point in their manuscript. I would also include a fractionation experiment to see whether ARF7 accumulates in different compartments (nucleus vs. cytoplasm).

R: Usage of different loading controls can achieve a certain level of "religious" argumentation by the scientific community. We understand the reviewer's concerns, but we respectfully disagree for several reasons:

A) Our data is provided as relative **ratios of ARF7 abundance normalized to input** (RuBisCO L). Even if RuBisCO L accumulates in *atg* mutants, that is still factored in the quantifications (ratio) and therefore, **loading differences should be accounted for**. In addition, those **ratios are averages of several blots** (available as source data). Analyses of those blots show that the differences in loading between Col-0 and the *atg* mutants are not an issue (2 examples provided below). On top of this, the argumentation by reviewer1 is contrary to their own hypothesis: if RuBisCO L accumulates in *atg* mutants, then *atg* mutant samples would “appear” to have more protein than they really do. Following this logic, **ARF7 levels accumulate to higher levels in *atg* mutants** than what we show in the blots, **making our hypothesis more convincing**, not less!

B) As seen below, we did try using tubulin for some of the blots in figure 1a. In those blots it can be seen that there is no major difference between tubulin and RuBisCO L as loading controls. However, **in our previous rebuttal, we provided a comparison of tubulin vs Ponceau S** staining of a whole membrane. That membrane was from one of the oscillation experiments (Figure 4A-E). There it was extremely easy to see that Tubulin does not reflect the overall protein content of the tissue. This is the advantage of using total protein staining vs an antibody against a single protein. However in contrast to tubulin, that same blot shows that **RuBisCO L is a better representative for the overall protein content**, matching the signal of the other bands seen with Ponceau. Because we did not want to use 2 loading controls, one for figure 1 and another for figure 4, and given that RuBisCO L is more consistent than Tubulin, we decided to keep RuBisCO as our only loading control. This is also supported by the fact that several methodological publications have shown that Ponceau S staining is a superior loading control than antibodies against specific housekeeping proteins like Actin and Tubulin (e.g. Gilda & Gomes 2013 Ann. Biochem; Salomao Fortes et al., 2016 Ann. Biochem; Bettencourt et al., 2021 Gene rep). Even further, publications like Nicot et al., 2005 J.Exp. Bot.; Bao et al., 2001 The Plant J.; Hausrat et al.,

2020 Develop Neuro.; Baojun et al., 2021 J. Cotton Res., and Chu et al., 2022, NPJ aging; show that it is not “just in our hands” that **tubulin is at best a debatable choice for control, displaying tissue and time dependent variability** both at the transcript and protein level in different organisms.

C) Ponceau s staining and RuBisCO L are one of the most widely used loading control in plant research, also in the autophagy field (Munch et al., 2014 Autophagy; Ustun et al 2018, TPC; Hafren et al., 2018 Plant Phys; Thirumalaikumar et al., 2020, Autophagy; Rodriguez et al., 2020., EMBOJ; Stephani et al., 2020 eLife; Shukla et al., 2021, Autophagy; Zhao et al., 2022, JCB; Qi et al., 2020 TPC). As a matter of fact, in Qi et al., 2020 it is possible to see that there is no difference in accumulation in RuBisCO L between Col-0 and *atg1* triple KO, making them comparable to actin. It is thus surprising to see the insistence of this reviewer against RuBisCO.

D) Autophagy has been shown to degrade several cytoskeleton components (e.g. Chen et al 2013., Toxicology; Rao et al., 2021 Autophagy) thus it is possible that autophagy also affects other components like Actin and Tubulin. Following this reviewer’s logic, they would be unsuitable as loading control.

E) Our microscopy data agrees with our western blots regarding the accumulation of ARF7 and shows precisely this is the case both in the nuclei and cytoplasm of the oscillation and mature zones.

For all these well-founded reasons, we trust our data and interpretations regarding ARF7 accumulation in *atg* mutants should be immaculate and no further experiments are needed.

Figure 1C: Since the authors claim NBR1 is degrading ARF7, it is essential to repeat this blot in the NBR1 mutant background.

R: We respectfully disagree with this comment. Firstly, this is a needless experiment which will bring little added value to our story as we already showed **that native ARF7 accumulates in *nbr1-c1* mutant** (figure 1A), and that NBR1 interacts with and co-localizes with ARF7 (figure 2 and 3). **We also emphasize this is the first time this experiment is requested.**

We also point at the contrast between the 2 rounds of reviews here: Previously this reviewer recommended but not demanded point 8: the crossing of ARF7 Venus to an *atg* mutant to further strengthen our data. The reviewer themselves did not deem that cross to be essential and mentioned that a downside was that the crosses could take long time to obtain (up to 6 months). We understood that cross would be important and we took the time to produce it and analyse it. Now, the same reviewer is recommending a cross to be essential (*nbr1-c1* x ARF7 venus), which will only provide redundant information at the expense of considerable delay. In our understanding this request is completely unnecessary.

As a side note, we are basically being asked to confirm western blots made against native ARF7 with a previously published antibody, by using transgenic tagged version ARF7. There is very little scientific merit to this request, and we fail to see how the community would accept this type of western blot confirmation becoming mandatory.

General comment: As F box protein AFF1 regulates ARF7 accumulation and nucleo-cytoplasmic partitioning it is still not clear to me why autophagy would need to regulate protein abundance of ARF7- is it only for the cytoplasmic pool? However, the image analysis shows that the nuclear signal is also increased. At this point I suggest performing fractionation experiments to reveal whether protein levels of ARF7 are altered in different compartment comparing *atg* mutants (and *nbr1*) with Col-0. I also suggest monitoring whether ARF7 is still degraded by selective autophagy in *aff1* mutant background. It could be also possible that AFF1 plays a role in its autophagic degradation. Considering previous findings on AFF1-ARF7 it essential to include a more detailed analysis to decipher the fate of ARF7.

R: We fail to see a lack of clarity regarding the role of autophagy in the clearance of ARF7. The role of AFF1 as a putative F-Box protein for ARF7 is consistent with autophagic and proteasome degradation, namely by promoting ubiquitination of ARF7 to facilitate its degradation by either protein turnover mechanisms. Jing et al 2022 explicitly states that “...Thus, it seems likely that multiple mechanisms exist to regulate ARF protein accumulation...”, opening themselves room for autophagic degradation of ARF7.

Moreover, the role of AFF1 in ARF7 regulation is not defined yet: **Jing et al., 2022 did not show any evidence that AFF1 ubiquitinates ARF7 or directly participates in ARF7s turnover!** In fact, those authors write in their discussion “*The most plausible explanation for this data is that SCF^{AFF1} mediates monoubiquitylation, or perhaps a ubiquitin chain that does not promote degradation, of ARF7 and ARF19 that affects its nucleo-cytoplasmic partitioning, possibly by blocking a nuclear export signal similar to what is seen in transcription factors such as p53.* And furthermore, in their *in vitro* degradation assay (figure 4B) they still see significant decrease of ARF7 levels in the Δ F-box-AFF1 samples.

It should be noted that also at the phenotypic level there are still points of contention regarding *aff1* mutants. Namely, the massive cytoplasmic accumulation and nuclear depletion of ARF1, ARF7, ARF19 (and potentially other ARFs) in *aff1* mutants should be linked to severe impairment in phenotypes dependent on these ARFs. Surprisingly, *aff1* mutants display normal root growth and only show mild insensitivity to auxin. Even more

surprising is the fact that those mutants were able to form LR, which are not seen in the *arf7 arf19* double mutant (Okushima et al 2007 TPC) and we duly discuss this (L506-510).

In sum, we have sufficiently proved that autophagy mediates ARF7 degradation and there are phenotypical consequences to ARF7 dependent developmental features. On the opposite, the role of AFF1 in ARF7 regulation is still far from clarified both at the mechanistic and phenotypical level. With that in mind, experiments looking at interplay of AFF1 and autophagy could be interesting but would require that the role of AFF1 in ARFs regulation is further clarified and we believe Dr. Straders group will do an excellent job at deciphering that in the future. Given the lack of clarity regarding AFF1 regulation of ARF7, we also think is difficult to expand our discussion with potential connections between AFF1 and autophagy in ARF7 turnover.

Figure 2: Since the authors state that ARF7 harbours AIMs, I strongly recommend mutating them and repeating interaction assays with ATG8 to strengthen their findings. AlphaFold-Multimer prediction might be very helpful for this. For what does ARF7 need NBR1, if ARF7 has functional AIMs and can interact with ATG8?

R: ARF7 has 14 predicted AIMs, mutating all of them and testing interactions would be extremely time consuming, and while interesting it's more suited for subsequent studies.

Did the authors test if ATG8 or NBR1 are present in the IPs in A and B? It would be more convincing and also consistent with their mcBiFC in 2C.

R: We have not tested this, and we do not understand how this experiment would make our data more convincing. Our data regarding ATG8 and NBR1 regulation of ARF7 is consistent with the extremely well documented role of ATG8 as an adaptor for NBR1. This request would add very little value to our story, but significantly delay publication.

Since NBR1 binds to ubiquitinated proteins, it would be important to show that ubiquitinated ARF7 is interacting with NBR1. This could be done e.g. in the presence of MG132 (blot in rebuttal letter). Additionally, to understand if AFF1 plays a role in ARF7 degradation by autophagy I suggest performing these experiments in the *aff1* mutant background.

R: We note that besides UBA domain, NBR1 also has AIM and PB1 domains, therefore interaction with ARF7 could occur in a ubiquitin independent matter. While interesting, this information is: a) not necessary to validate the core message of our story, b) would significantly delay publication; c) is more suited for future publications. As for AFF1, for the reasons explained above, we believe that these experiments are not relevant to our story.

Comment to mcBiFC: The authors show that the interaction of NBR1-ATG8-ARF7 also occurs in the nucleus. How is ARF7 then "extracted" from the nucleus and degraded by autophagy? It appears to be rather complicated...Or is this an artefact of the mcBiFC method.

I also strongly recommend strengthening the findings by in vitro assays with NBR1 (K11 mutant)-ARF7 and ATG8.

R: We do not understand the alleged complexity in this context. JOKA2 (NBR1 homologue in *N. benthamiana*) possesses nucleo-cytoplasmic localization, which is also true for their mammalian homologues p62 and NBR1 (Zientara-Rytter et al 2011, Autophagy; Huang et al 2021, Nature Comm). ATG8/LC3 is also known to be nuclear localized (e.g. Shim et al., 2020 Autophagy). Moreover, autophagy participates in degradation of nuclear components (see review Li and Nakatogawa 2022, Trends Cell Bio), which is consistent with our data. At

any rate, we think that while interesting, this is not immediately relevant to the validity of our core story and is more suited for subsequent studies.

Additionally, I would repeat all interaction data in the presence or absence of IAA (in the light of Figure 5) and also perform fractionations prior IPs to elucidate if the interaction is in the nucleus or cytoplasm.

R: While we believe the experiment suggested is interesting, this is more suited for a follow-up study.

Figure 4F: What about the expression of LBD16/33 as well as other auxin responsive genes in the *nbr1* mutant? If the authors want to provide evidence that NBR1 is degrading ARF7 they need to repeat this in the *nbr1* mutant. Instead of the proteomics in Figure 5, I would strongly suggest performing a transcriptomic experiment in *atg2* and *nbr1* (-/+IAA) as ARF7 is a transcription factor. If autophagy is essential for ARF7 turnover I would expect some changes in auxin responsiveness in the mutants. Otherwise, I would suspect that the phenotypes presented in Figure 6 are due to the pleiotropic nature of *atg* mutants (see comments below).

R: We disagree with this comment and respectfully decline performing further experiments for the following reasons: a) There should not be an “IF” in what concerns ARF7 degradation via NBR1-mediated autophagy; we provided robust evidence of that in this paper by different complementary methods. b) We already proved that *LBD16* accumulates in *atg2-1* (figure 4F); c) we also have previously (Rodriguez et al 2020) and in this paper (Figures 5 and 6) that autophagy mutants respond to auxin differently than control plants. d) While we show increase in *LBD16* levels in *atg2-1* this would not necessarily even have to be the case, as *LBD16* could be under different type of compensatory mechanisms and not accumulate despite enhanced ARF7 levels in *atg* mutants. As for the comments about pleiotropism, we discuss that possibility extensively in the manuscript already but we also point at the fact that misregulation of a key LR regulator like ARF7 should be an important contributor to the phenotype in question (L525-543).

Figure 4G: To me it is not clear why this experiment was not performed in *atg* mutants. Pepstatin is inhibiting aspartyl proteases, and this assay only demonstrates that they seem to play a role in this. It is essential to perform this experiment in autophagy deficient mutant as well as in *nbr1*.

R: We agree that would have been optimal to do this experiment in *atg* mutants. However, it should be noted that this experiment was performed in reply to a request by a past editor and thus made in a short time window (i.e. one month). As this reviewer mentioned in the first round of review, crosses can take over 6 months (or even more given *atg* mutants have deficiencies in gamete production) and therefore it would have been impossible to perform a cross between *atg2-1* and *DR5::Luc* in a timely fashion. We also remind this reviewer that chemical inhibition of vacuolar proteases is an accepted and often used resource to study autophagy impairment, even beyond the plant field (Li et al., 2013 JBC; Tanida et al., 2005; Autophagy; Saiki et al., 2011 Autophagy; Duan et al., 2015 Arch. Vir; Zhao et al 2019., Cell Death & Diff; Subramanian et al., 2021 Nat. Comms).

I cannot find any description of Figure 4J-L in the results section.

R: We thank the reviewer for this comment, we forgot to add it to the text (line 322-324).

Figure 5: NAA also has other effects, e.g. on endocytosis and the endomembrane system (Narasimhan et al., 2021, Plant Phys). Thus, the authors should repeat their experiments using IAA (Figure 5A-C).

R: NAA has been widely used as a synthetic version of auxin for many decades (e.g. Himanen et al., 2002 TPC; Okushima et al., 2007 TPC; Brunoud et al., 2012 Nature; Orosa-Puente et al., 2018 Science; Meier et al., 2020; Nat. Plants; Taylor et al., 2021 PNAS; Kircher & Schopfer 2023, Cur. Biol.). While it's indeed true that NAA has other effects, the fact that NAA primary usage is that of a "similar but more potent" version of IAA is not opened to dispute and therefore should be perfectly acceptable to in the context of our experiments.

5E/F to me it is not really clear what is up-/downregulated - do the authors only look into auxin responsive genes? Since ARF17 is a transcription factor it is more essential to look into the rapid transcriptional response (see comment above).

R: We respectfully disagree with this. The aim of this experiment was to evaluate how proteome remodelling upon rapid NAA treatment was affected in *atg* mutants. Performing transcriptomics in this context would be irrelevant to our story; autophagy's impact on protein content changes should be direct but its impact at transcriptional level should be indirect. Moreover, transcript and protein abundance are not well correlated (reviewed in Vélez-Bermúdez and Schmidt 2014 and our own data on ARF7 demonstrates that).

Figure 6: The authors claim in the title and abstract that ARF7 is degraded by NBR1 selective autophagy but do not show any physiological data in Figure 6: How does NBR1 respond to auxin? What about LR formation in *nbr1* and other auxin-related phenotypes? It is not surprising and known that autophagy deficient mutants have developmental defects. Thus, it is more important that the *nbr1* mutant displays these phenotypes as autophagy mutants are more pleiotropic.

R: We respectfully disagree with this comment. Firstly, we demonstrated that NBR1 participates in the degradation of ARF7, therefore, we find the comment reductive to the data we have presented. As for pleiotropism, we have duly discussed that possibility the manuscript (L525 to 543). With that said, we iterate that given the central role of ARF7 to LR formation, it is extremely unlikely that misregulation of ARF7 in *atg* mutants would not be partly responsibly, if not chiefly responsible, for the LR defects in *atg* mutants, and our data support this conclusion.

Still, we executed the experiment requested (new figure EV5) and unequivocally demonstrate that *NBR1* KO lines produce significantly less LRs than WT plants and this is not restored by NAA supplementation, proving these reviewers point mute.

Referee #2:

The authors have satisfactorily addressed my comments and concerns in their revised manuscript, resulting in notable improvements to the document.

R: We thank the reviewer for this comment and appreciation of our efforts.

Dear Eleazar,

Thank you for the submission of your revised manuscript to EMBO Reports. Before we can proceed with the official acceptance of your study, I note a few things that we need from the editorial side:

- Please provide the manuscript as a .docx file without the figures.
- Please add a 'Disclosure and competing interests statement'. For more information see <https://www.embopress.org/page/journal/14693178/authorguide#conflictsofinterest>
- The manuscript sections should be in the following order:
Title page - Abstract & Keywords - Introduction - Results - Discussion - Methods - Data Availability - Acknowledgments - Disclosure Statement & Competing Interests - References - Figure Legends - Tables with legends - Expanded View Figure Legends.
- Please add up to 5 keywords.
- There is a discrepancy in the way the authors are listed in the manuscript file (last name first name initial) and the online system (first name last name). Please list the author's full names on the manuscript.
- Regarding the Author Contributions, we now use CRediT to specify the contributions of each author in the journal submission system. Therefore, please remove the Author Contributions from the manuscript file and make sure that the author contributions in our manuscript tracking system are correct and up-to-date. The information you specified in the system will be automatically retrieved and typeset into the article. You can enter additional information in the free text box provided, if you wish.
- Funding information is retrieved from the information entered in the manuscript tracking system so it is important that this information is complete. In this regard, we note that the Novo Nordisk Foundation (NNF) (grant agreement number NNF19SA0059305) is missing. Please add this funder/grant.
- Data availability section: please insert a link that resolves directly to the dataset PXD042834 at PRIDE (not just to PRIDE itself) and please do not forget to remove the reviewer access from the text.
- Source data: please add a link in the Data availability section that resolves to the dataset on BiolImage archive.
- The Source data folders for EV figures need to be grouped and uploaded as one zipped folder. This means that you put all current EV Source data folders into one folder called Source Data EV figures and then zip and upload this folder.
- Supplementary Table 1 (primers) needs to be renamed to Table 1.
- The scale bars in Figure 5A appear very thin and might not be well visible at final size.
- Our production/data editors have asked you to clarify several points in the figure legends (see below). Please incorporate these changes in the manuscript and return the revised file with tracked changes with your final manuscript submission.
 - A) Please note that a separate 'Data Information' section is required in the legends of figures 1a-c.
 - B) Please note that in figures 5d; 6b; there is a mismatch between the annotated p values in the figure legend and the annotated p values in the figure file that should be corrected. Figure 5d misses a definition of **** and Figure 6b misses the definition of ** and ***.
 - C) Please note that information related to n is missing in the legends of figures 3b; 4f; EV 4b. Although 'n' is provided, please describe the nature of entity for 'n' in the legend of figure 4f.
 - D) Please note that the error bar is not defined in the legend of figure 1b.
 - F) Please note that the scale bar needs to be defined for figures 1d-e; 4j.
- The paragraphs on 'Sample preparation for proteomic analysis' and 'Data acquisition by LC-MS' in the methods seem to match the corresponding methods paragraphs in Bastrup J et al, 2022 (PMID 34929167). It is fine not to reinvent the wheel when describing such procedures but I suggest citing Bastrup et al.
- I have introduced two minor changes to the Abstract, which I suggest to incorporate (e.g. show instead of prove). My suggestion is below my signature.
- Finally, EMBO Reports papers are accompanied online by A) a short (1-2 sentences) summary of the findings and their significance, B) 2-3 bullet points highlighting key results and C) a synopsis image that is 550x300-600 pixels large (width x

height) in PNG for JPG format. You can either show a model or key data in the synopsis image. Please note that the size is rather small and that text needs to be readable at the final size. Please send us this information along with the revised manuscript.

- On a different note, I would like to alert you that EMBO Press offers a new format for a video-synopsis of work published with us, which essentially is a short, author-generated film explaining the core findings in hand drawings, and, as we believe, can be very useful to increase visibility of the work. This has proven to offer a nice opportunity for exposure i.p. for the first author(s) of the study. Please see the following link for representative examples and their integration into the article web page:

https://www.embopress.org/video_synopses
<https://www.embopress.org/doi/full/10.15252/emj.2019103932>

Kind regards,

Martina

Auxin dictates root architecture via the Auxin Response Factor (ARF) family of transcription factors, which control lateral root (LR) formation. In Arabidopsis, ARF7 regulates the specification of prebranch sites (PBS) generating LRs through gene expression oscillations and plays a pivotal role during LR initiation. Despite the importance of ARF7 in this process, there is a surprising lack of knowledge about how ARF7 turnover is regulated and how this impacts root architecture. Here, we show that ARF7 accumulates in autophagy mutants and is degraded through NBR1-dependent selective autophagy. We demonstrate that the previously reported rhythmic changes to ARF7 abundance in roots are modulated via autophagy and might occur in other tissues. Additionally, we show that the level of co-localization between ARF7 and autophagy markers oscillates and can be modulated by auxin to trigger ARF7 turnover. Furthermore, we observe that autophagy impairment prevents ARF7 oscillation and reduces both PBS establishment and LR formation. In conclusion we report a novel role for autophagy during development, namely by enacting auxin-induced selective degradation of ARF7 to optimize periodic root branching.

Rev_Com_number: RC-2022-01645
New_manu_number: EMBOR-2023-58615V2
Corr_author: Rodriguez
Title: NBR1-mediated selective autophagy of ARF7 modulates root branching

All editorial and formatting issues were resolved by the authors.

Dr. Eleazar Rodriguez
University of Copenhagen
Biology
Ole Maaloees Vej 5
Copenhagen, orcid||||| 2200
Denmark

Dear Eleazar,

I am very pleased to accept your manuscript for publication in the next available issue of EMBO reports. Thank you for your contribution to our journal.

Kind regards,

Martina
